# Serotonin neuromodulation directs optic nerve regeneration

Kristian Saied-Santiago, Melissa Baxter, Jaffna Mathiaparanam and Michael Granato*

## ABSTRACT

Optic nerve (ON) regeneration in mammalian systems is limited by an overshadowing dominance of inhibitory factors. This has severely hampered the identification of pro-regenerative pathways. Here, we take advantage of the regenerative capacity of larval zebrafish to identify pathways that promote ON regeneration. From a small molecule screen, we identified modulators of serotonin (5-HT) signaling that inhibit ON regeneration. We find that several serotonin type-1 (5-HT1) receptor genes are expressed in retinal ganglion cells during regeneration and that inhibiting 5-HT1 receptors or components of the 5-HT pathway selectively impedes ON regeneration. We show that 5-HT1 receptor signaling is dispensable during ON development yet is required for regenerating axons to emerge from the injury site. Blocking 5-HT receptors once ON axons have crossed the chiasm does not inhibit regeneration, suggesting a selective role for 5-HT receptor signaling early during ON regeneration. Finally, we show that agonist-mediated activation of 5-HT1 receptors leads to enhanced and ectopic axonal regrowth. Combined, our results provide evidence for mechanisms through which serotonin-dependent neuromodulation directs ON regeneration *in vivo*.

KEY WORDS: Zebrafish, Optic nerve, Axon regeneration, Serotonin, 5HT

## INTRODUCTION

The optic nerve is considered part of the central nervous system (CNS) and is composed of retinal ganglion cell (RGC) axons and glial cells to relay visual information from the retina to the brain. Despite its essential role in vision, the mammalian optic nerve has limited regenerative capacity (Laha et al., 2017; Fischer and Leibinger, 2012). In response to injury or disease-induced damage, most RGCs die (Villegas-Pérez et al., 1993). Even when RGCs survive the initial insult, inhibitory factors severely restrict axonal growth, resulting in loss of visual function (Silver and Miller, 2004; Buckingham et al., 2008). Combinations of growth factors have been shown to enhance the regrowth of RGC axons (Li et al., 2016; Park et al., 2008; de Lima et al., 2012); however, these axons frequently fail to properly navigate the optic chiasm and only few reach their original brain targets (Pernet et al., 2013; Bray et al., 2017), suggesting that proper regeneration requires a more comprehensive understanding of the pathways that promote various aspects of axonal

Department of Cell and Developmental Biology, Perelman School of Medicine, University of Pennsylvania, Philadelphia, PA 19104, USA.

*Author for correspondence (granatom@pennmedicine.upenn.edu)

K.S., 0000-0002-8922-2683; J.M., 0000-0001-9405-9772; M.G., 0000-0003-3878-9468

regrowth, including pathfinding and target selection. Over the past decade, it has become clear that spontaneous regeneration of injured CNS axons is not limited to non-mammalian systems but is also operant in mammals such as the naked mole rat or the spiny mouse (Park et al., 2017; Nogueira-Rodrigues et al., 2022). In these spontaneous regeneration models, optic nerve regeneration likely proceeds due to the dominance of pro-regeneration pathways, and, hence, vertebrate model systems, including teleost fish, represent a unique opportunity to identify regeneration-promoting pathways without the overshadowing effects of inhibitory pathways (Becker and Becker, 2007). Moreover, larval zebrafish are optically transparent, which facilitates monitoring of optic nerve regeneration *in vivo* and in real time (Harvey et al., 2019). Finally, larval zebrafish are amenable to small molecule screens and thus represent an unparalleled opportunity to identify factors promoting optic nerve regeneration *in vivo*.

Serotonin (5-HT) is a neurotransmitter that binds to 5-HT cell-surface receptors to primarily mediate communication between neurons through synaptic connections (Lipton and Kater, 1989; Zhou and Hablitz, 1999). 5-HT receptors are subdivided into seven families (Barnes and Sharp, 1999) based on their protein structure, signal transduction, and pharmacology (Hoyer et al., 1994). Recently, vertebrate studies have identified 5-HT receptors as developmental regulators of CNS axonal growth (reviewed by Trakhtenberg and Goldberg, 2012). For example, 5-HT receptor signaling has been shown to play instructive roles in guiding commissural axons at the CNS midline (Xing et al., 2015). In rodents, the 5-HT1B receptors and monoamine oxidase, an enzyme that breaks down 5-HT, are required during development to properly sort retinal projections (Upton et al., 1999, 2002). Finally, serotonin receptor signaling has also been implicated in post-developmental processes, including axonal regrowth. Following a spinal cord transection in adult zebrafish, 5-HT1B receptors facilitate the regeneration of spinal interneurons (Huang et al., 2021). Additionally, serotonin modulates axonal regrowth through the 5-HT1A and 5-HT2 receptors in goldfish retinal explants after an optic crush injury (Lima et al., 1994; Schmeer and Lima, 2000; Schmeer et al., 2001). Yet despite their function in modulating axonal growth and guidance, the *in vivo* requirement of 5-HT signaling in CNS axonal regeneration, including optic nerve regeneration, has remained elusive.

We recently developed an optic nerve transection assay in larval zebrafish that allows for live examination of the neurogenesis-independent process of spontaneous optic nerve regeneration (Harvey et al., 2019). Here, using this *in vivo* assay, we conducted a screen using a small molecule library (Selleckchem Bioactive) against defined molecular targets. From this screen, we identified four compounds modulating serotonin signaling that impair optic nerve regeneration. Focusing on two of the identified small molecules, an agonist and an antagonist of 5-HT1 receptors, we provide compelling evidence that 5-HT1 signaling promotes axonal growth during the early stages of optic nerve regeneration. We show that blocking 5-HT1 receptors during optic nerve development does not impair developmental axon growth, arguing that 5-HT receptor signaling plays a selective role during regeneration. Lastly, we find that ectopic

activation of 5-HT1 receptors leads to enhanced but also misguided regrowth of optic nerve axons, suggesting that 5-HT receptor-dependent neuromodulation plays an instructive role in directing regenerating optic nerve axons toward their original brain targets.

## RESULTS

### A small molecule screen uncovers components of the 5-HT signaling pathway as modulators of RGC axon regeneration

To identify signaling pathways that promote RGC axon regeneration *in vivo*, we screened a small molecule library consisting of over 1400 FDA-approved compounds. These compounds cover a broad range of identified targets and over 40 genetic signaling pathways as defined by PANTHER pathway analysis (Mi and Thomas, 2009), including agonists and antagonists targeting growth factor pathways, neuronal cell-surface receptors, and neurotransmitter systems (Table S1). To test their *in vivo* roles in CNS regeneration, these compounds were applied to larvae containing the *Tg(isl2b:GFP)* transgene, which labels most, if not all, RGC neurons and their axons express GFP (Fig. 1A,A′) (Pittman et al., 2008). Injury was induced when larvae were at 5 days post-fertilization (dpf), a time point when their visual system is already functional (Brockerhoff et al., 1995; Easter and Nicola, 1996). In these animals, we fully transected the optic nerve halfway between the retinal exit point and the optic chiasm using a sharpened tungsten needle. We previously showed that regrowing RGC axons innervate both ipsilateral and contralateral tecta following complete axonal transection (Fig. 1B) (Harvey et al., 2019). Therefore, to simplify analysis and ensure the regenerating axons from a single injured nerve were scored properly, we removed the uninjured right eye (Fig. 1B′). After transection, the transected optic nerve undergoes a stereotypic timeline of recovery. By 24 h post-transection (hpt), the nerve segment distal to the injury site had undergone fragmentation/degeneration, and regenerating optic nerve axons emerged from the nerve stump and began to extend toward the optic chiasm located at the CNS midline (Fig. 1C,C′) (Harvey et al., 2019). At 32 hpt, a small group of regenerating RGC axons had reached the midline (Fig. 1D), and by 48 hpt regrowing RGC axons in wild-type larvae had extended past the optic chiasm and begun to re-innervate the peripheral edges of the optic tecta (Fig. 1E,E′). In order to assess the *in vivo* effects of the small molecule library, particularly on early axon guidance and extension, we applied small molecules (or a DMSO control) to larvae 24 h after full transection of the optic nerve (Fig. 1C′). From the small molecule library, we identified and independently confirmed four small molecules known to modulate serotonin signaling that resulted in impaired optic nerve regeneration. One identified molecule is the endogenous ligand serotonin, while another, tranylcypromine, is an antagonist that inhibits the monoamine oxidase and the lysine-specific demethylase 1 (LSD1) enzymes (Table 1). The remaining two molecules represent agonists and antagonists against 5-HT1 receptors. Specifically, WAY-100635 is a 5-HT antagonist that inhibits type-1A receptors, while zolmitriptan has been validated as an agonist that activates 5-HT type-1B/1D receptors (Table 1). Given this selectivity, we focused on further determining the roles of 5-HT1 receptors in optic nerve regeneration.

### Inhibiting 5-HT1 receptors significantly reduces RGC regenerative axonal growth

We first characterized the optic nerve regeneration phenotype observed after treatment with the small molecule WAY-100635. This serotonin receptor antagonist is a well-characterized inhibitor of 5-HT1A receptors *in vitro* (Ferreira et al., 2010) and *in vivo* (Maximino et al., 2013; Long et al., 2023), with 100-fold higher binding selectivity to 1A receptors over other subtypes (Forster

et al., 1995) and which shows lower affinity to dopamine D4 receptors (Chemel et al., 2006) than 5-HT1 receptors. We exposed larvae with fully transected optic nerves to a vehicle (0.3% DMSO) or WAY-100635 from 24 to 48 hpt, and at 48 hpt quantified the number of transected optic nerves that had extended to the contralateral tectum. Compared to DMSO-treated control animals, we observed a significant and dose-dependent reduction of optic nerve regrowth in WAY-100635-treated larvae (Fig. 2A-D, compare DMSO controls with 5 µM or 50 µM WAY-100635-treated larvae). Specifically, after incubating larvae in 50 µM WAY-100635 media, we observed a 2.2-fold increase of nerves (Fig. 2D, Fig. S1B) that stalled at various positions along their regenerative path (Fig. 2B,C, yellow arrowheads in chiasm panels) and failed to re-innervate the contralateral tectum (Fig. 2B,C, white dashed lines in tecta panels). To determine whether RGC axons in animals treated with 50 µM WAY-100635 stall before or after the optic chiasm, we examined whether optic nerves failed to cross the optic chiasm by 48 hpt. Compared to control animals, regenerating optic nerves in WAY-100635 treated larvae failed to cross the optic chiasm at a significantly higher rate (42% in 50 µM WAY-100635-treated larvae compared to 19% in control nerves; Fig. S1A, black bars). Lastly, to determine whether the growth defects observed in optic nerves of WAY-100635-treated animals were due to regenerating axons deviating from their pre-transection trajectory or stalling prematurely along their trajectory path, we quantified the number of optic nerves displaying ectopic axonal regrowth (Fig. 2B, yellow arrow). After treatment with low (5 µM) or high (50 µM) concentrations of the 5-HT1 antagonist WAY-100635, we failed to observe a significant increase in ectopic axonal regrowth (Fig. 2E, Fig. S1C), consistent with the idea that WAY-100635 causes RGC axons to stall before or at the optic chiasm. Combined, these results strongly suggest that 5-HT1 receptors promote RGC axonal regrowth.

To determine whether 5-HT1 receptors promote RGC axonal regrowth along the entire regenerative trajectory to the optic tectum or selectively at distinct portions of the regenerative path, we blocked 5-HT1 receptors once RGC axons had reached the optic chiasm. For this, we applied the antagonist WAY-100635 to larvae with fully transected optic nerves at 32 hpt, and assessed contralateral tectal re-innervation at 48 hpt (Fig. 2F, top). We found that optic nerves in larvae treated with 50 µM WAY-100635 re-innervated the contralateral tectum to a similar extent as that observed in DMSO controls (Fig. 2F, bottom; Fig. S1D). Together with our findings that inhibition of 5-HT1 receptors at 24 hpt significantly reduces the re-innervation of RGC axons to the contralateral tecta (Fig. 2D), these results provide compelling evidence that 5-HT1 receptors promote RGC axon regrowth during the early stages of axon regeneration once axons emerge from the nerve stump and extend towards the optic chiasm.

We next examined whether 5-HT1 receptor signaling acts to promote axonal regrowth broadly within the CNS or whether it selectively promotes optic nerve regeneration. For this, we examined the role of 5-HT1 receptors in spinal cord regeneration. Specifically, we focused on a pair of reticulospinal neurons, the Mauthner cell neurons, to quantitatively and at single-neuron resolution determine the effects of 5-HT1 receptor antagonists in spinal cord regeneration. A single Mauthner cell neuron resides on either side of the hindbrain, and its axon extends from the hindbrain to the tip of the spinal cord (Eaton et al., 1977). We used a well-established assay to laser-transect Mauthner axons in 5 dpf larvae (Bremer et al., 2019) to test whether blocking 5-HT1 receptors impairs Mauthner axonal regeneration. We added 50 µM WAY-100635 at 1 hpt and then replaced the WAY-100635-containing incubation media every 24 h until 72 hpt (Fig. 2G, top). We found that, compared to DMSO-treated control

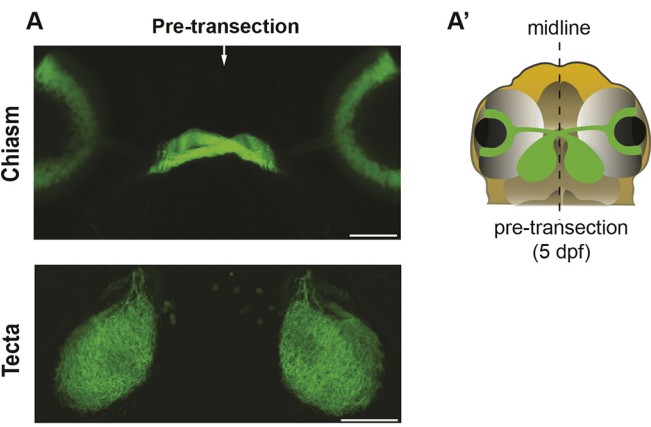

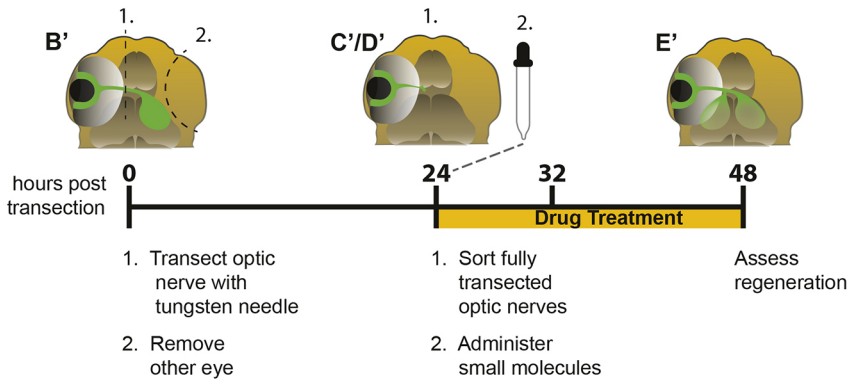

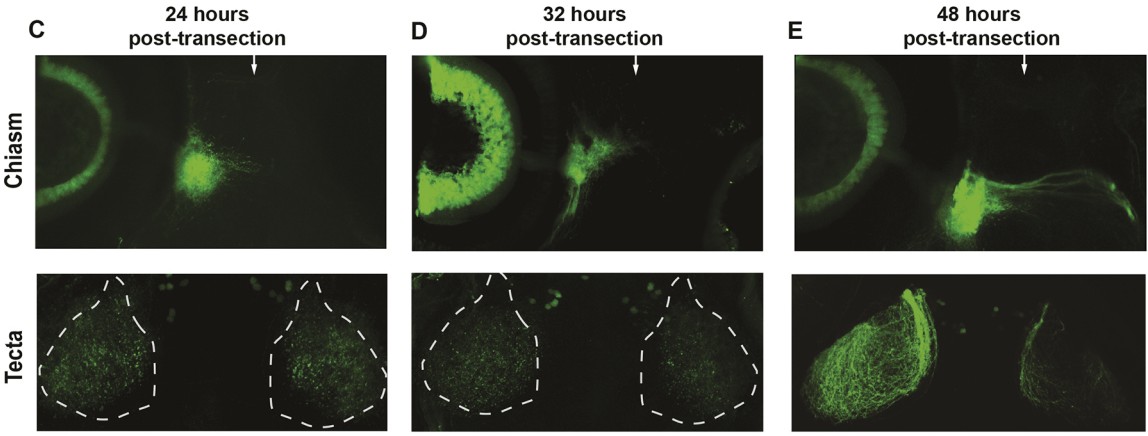

**Fig. 1. Regenerating RGC axons robustly regrow to the optic tecta by 48 h post-transection.** (A-E) Fluorescent representative images of *Tg(Isl2b:GFP)* fixed larvae visualized using confocal imaging at distinct time points: pre-transection, 24 h post-transection, 32 h post-transection and 48 h post-transection. Arrows depict the midline at the optic chiasm. Different larvae were used across time points. Scale bars: 50 µm. (A,A′) At 5 dpf, RGC axons have crossed the midline at the optic chiasm (top) and innervated the contralateral tecta (bottom). (B) Timeline of optic nerve transection, regeneration and larvae exposure to small molecules. Schematics describe the pre-transection and post-transection (0, 24, 32 and 48 hpt) time points during optic nerve regeneration and its assessment in larval zebrafish. (B′) At 0 hpt, the left optic nerve of *Tg(Isl2b:GFP)* larvae was transected, and the right eye was removed as detailed in the Materials and Methods. (C,C′) At 24 hpt, transected axons begin to extend towards the optic chiasm (chiasm region; C, top). Complete transection of the optic nerve was corroborated by examining larvae dorsally (tecta region; C, bottom) at 22-24 hpt for remnants of Isl2b:GFP nerve structures. Degenerated tecta are outlined with dashed lines. (D,D′) At 32 hpt, a small group of regeneration RGC axons have reached the midline (D, top). Degenerated tecta are outlined with dashed lines (D, bottom). (E,E′) At 48 hpt, wild-type larvae regrow their RGC axons toward the optic chiasm (E, top) and begin re-innervating the peripheral edges of the tecta (E, bottom). Live zebrafish larvae treated with small molecules at 24 hpt were assessed to determine whether optic nerve regeneration was impaired. Top and bottom panels for each time point are of the same larva. At 48 hpt, the ipsilateral tectum (left tectum) is typically more innervated than the contralateral tectum. Images are representative of at least six different samples per time point.

**Table 1. Small molecules targeting the 5-HT signaling pathway impair optic nerve regeneration**

| Small molecule | Type of small molecule | Target(s) | Phenotype |
|---|---|---|---|
| WAY-100635 | Antagonist | 5-HT1A receptor | Reduced tectal re-innervation |
| Zolmitriptan | Agonist | 5-HT1B/1D receptor | Increased ectopic regrowth |
| Tranylcypromine | Antagonist | Monoamine oxidase/LSD1 | Reduced tectal re-innervation |
| Serotonin | Agonist | Multiple 5-HT receptors | Reduced tectal re-innervation |

animals, the length of Mauthner axonal regrowth at 96 hpt in larvae treated with the 5-HT1 antagonist WAY-100635 was indistinguishable (Fig. 2G, bottom), demonstrating that Mauthner cell axon regeneration is unaffected by the presence of 5-HT1 receptor inhibitors. Combined with the effect of 5-HT1 receptor signaling on RGC regeneration, our results suggest a selective role for 5-HT1 receptors in RGC axon regeneration.

### 5-HT1 receptor genes are expressed in RGC neurons during RGC axon regeneration

5-HT1 receptors are expressed in the vertebrate CNS, including in the retina of mice and zebrafish larvae (Upton et al., 1999; Norton et al., 2008). Moreover, an existing single-cell transcriptomics dataset used to generate the molecular taxonomy of RGCs in zebrafish showed that 5-HT1A receptors are expressed in a subset of RGCs in the 5 dpf zebrafish larvae (Kölsch et al., 2021). Given the apparent selective role of 5-HT1 receptors in RGC axonal regeneration, we wondered whether 5-HT1 receptors are expressed in RGC neurons during regeneration. To detect mRNA expression of 5-HT1 receptors in the RGC layer, we performed whole-mount fluorescence *in situ* hybridization (FISH) with hybridization chain reaction (HCR) (Choi et al., 2018). We obtained specific probes for the four zebrafish 5-HT1 receptors targeted by small molecules that impaired optic nerve regeneration from our screen: *htr1aa*, *htr1ab*, *htr1b* and *htr1d*. After incubating *Tg(isl2b:GFP)* 5 dpf larvae with a probe mix against all four *htr1* receptors prior to optic nerve transection, we detected expression of these serotonin receptors distributed across the RGC layer [compare Fig. 3A-C (no probe) with Fig. 3D-F (four *htr1* receptors)] and in Isl2b:GFP RGC neurons (Fig. 3F′, yellow dashed circles). Next, we incubated uninjured larvae or larvae that were fixed at 24 and 32 h after transecting both of their optic nerves with a probe mix against *htr1aa* and *htr1ab*, receptors targeted by the antagonist WAY-100635. We detected expression of these receptors in RGC neurons before transection [Fig. 3G-I and yellow dashed circles in Fig. 3I′ (uninjured)] and also during regeneration at 24 hpt [Fig. 3J-L and yellow dashed circles in Fig. 3L′ (transected)]. However, we did not observe any staining after incubating transected larvae with probes against these receptors during regeneration at 32 hpt (compare Fig. S2A-C and Fig. S2D-F). Thus, 5-HT1A receptor genes are expressed in RGC neurons at a critical time when 5-HT1 receptors promote RGC axonal regrowth towards the optic chiasm.

### 5-HT1A receptor signaling is dispensable for developmental RGC axonal growth

Differentiation of RGC neurons begins around 28 h post-fertilization (hpf), and soon thereafter, RGC axons exit from the retina and extend to the optic chiasm (Laessing and Stuermer, 1996). By 48 hpf, the optic fissure closes, and most RGC axons

have arrived at the optic tectum (Fig. 4A) (Stuermer, 1988). Expression of 5-HT1 receptor genes has previously been detected in the developing retina and tectum of zebrafish (Pei et al., 2016). Since 5-HT1 receptors are expressed in RGC neurons in 5 dpf larvae, we examined whether 5-HT1 receptor-dependent signaling is also required to promote RGC axonal growth during development. Considering the developmental timeline of RGCs, we began treating embryos with the antagonist WAY-100635 at 24 hpf. We then examined whether developing optic nerves had reached the optic tecta at 48 hpf (Fig. 4A). Tectal innervation of optic nerve axons in WAY-100635-treated embryos was indistinguishable compared to DMSO-treated controls (Fig. 4B-D). Therefore, we conclude that 5-HT1A receptor function is likely dispensable in development for RGC axons to project to the optic tectum.

### 5-HT1 receptors direct regenerating RGC axons towards the optic chiasm

We next asked whether 5-HT1 receptor signaling plays an instructive role during optic nerve regeneration. To test this, we used the 5-HT1B/1D selective agonist zolmitriptan (Wurch et al., 1997; de Almeida et al., 2001), with modest affinity to 5-HT1A receptors, to exogenously activate 5-HT1 receptors during optic nerve regeneration and measured its effects on regenerating RGC axons. The *htr1b* and *htr1d* receptors are expressed in RGC neurons prior to optic nerve transection and during regeneration at 24 hpt [Fig. S3A-F and see mRNA expression in Isl2b:GFP neurons (yellow dashed lines) in Fig. S3C′,F′]. Following optic nerve transection, we exposed larvae to zolmitriptan from 24 to 48 hpt to induce 5-HT1 receptor activation and assessed regeneration at 48 hpt. Compared to DMSO-treated control animals (Fig. 5A), treatment with 5 µM zolmitriptan significantly increased the number of optic nerves that re-innervated the contralateral tectum (Fig. 5C, Fig. S4A; 1.6-fold change increase, compare DMSO controls with 5 µM zolmitriptan-treated larvae) without inducing significant ectopic regrowth (Fig. 5D, Fig. S4B), demonstrating that exogenous activation of 5-HT1 receptors enhances RGC axonal regeneration. However, increasing the agonist concentration to 50 µM caused a 2.3-fold increase of optic nerves that exhibited significant ectopic axonal regrowth (Fig. 5D; compare DMSO controls with 50 µM zolmitriptan-treated larvae). Specifically, in zolmitriptan-treated larvae, the growth of regenerating RGC axon fascicles became misguided from their pre-transection trajectory soon after exiting the injury site (Fig. 5B,B′, yellow arrowheads). In addition, in a small fraction (11%) of zolmitriptan-treated animals, regenerating RGC axons initially extended towards the optic chiasm but then changed their course and continued on ectopic routes, ultimately failing to innervate the contralateral tectum (Fig. 5B, white arrowhead). Lastly, exposure of larvae to 50 µM zolmitriptan did not affect the capacity of most regenerating axons to cross the optic chiasm and innervate the contralateral tectum (Fig. 5C; no fold-change when comparing DMSO controls with 50 µM zolmitriptan-treated larvae). Thus, treatment with 5-HT1 receptor agonists at low concentrations (5 µM) enhances tectal re-innervation without causing significant ectopic regrowth, while high concentrations (50 µM) lead to severe anterior-posterior axonal misguidance. Combined, our results agree with the idea that *in vivo* 5-HT1 receptors can direct regenerating optic nerve axons, consistent with an instructive role. Finally, forced activation of 5-HT1 receptors at 32 hpt failed to elicit significant defects in optic nerve regeneration, including ectopic regrowth (Fig. S3G,H), further underscoring the notion that serotonin signaling modulates

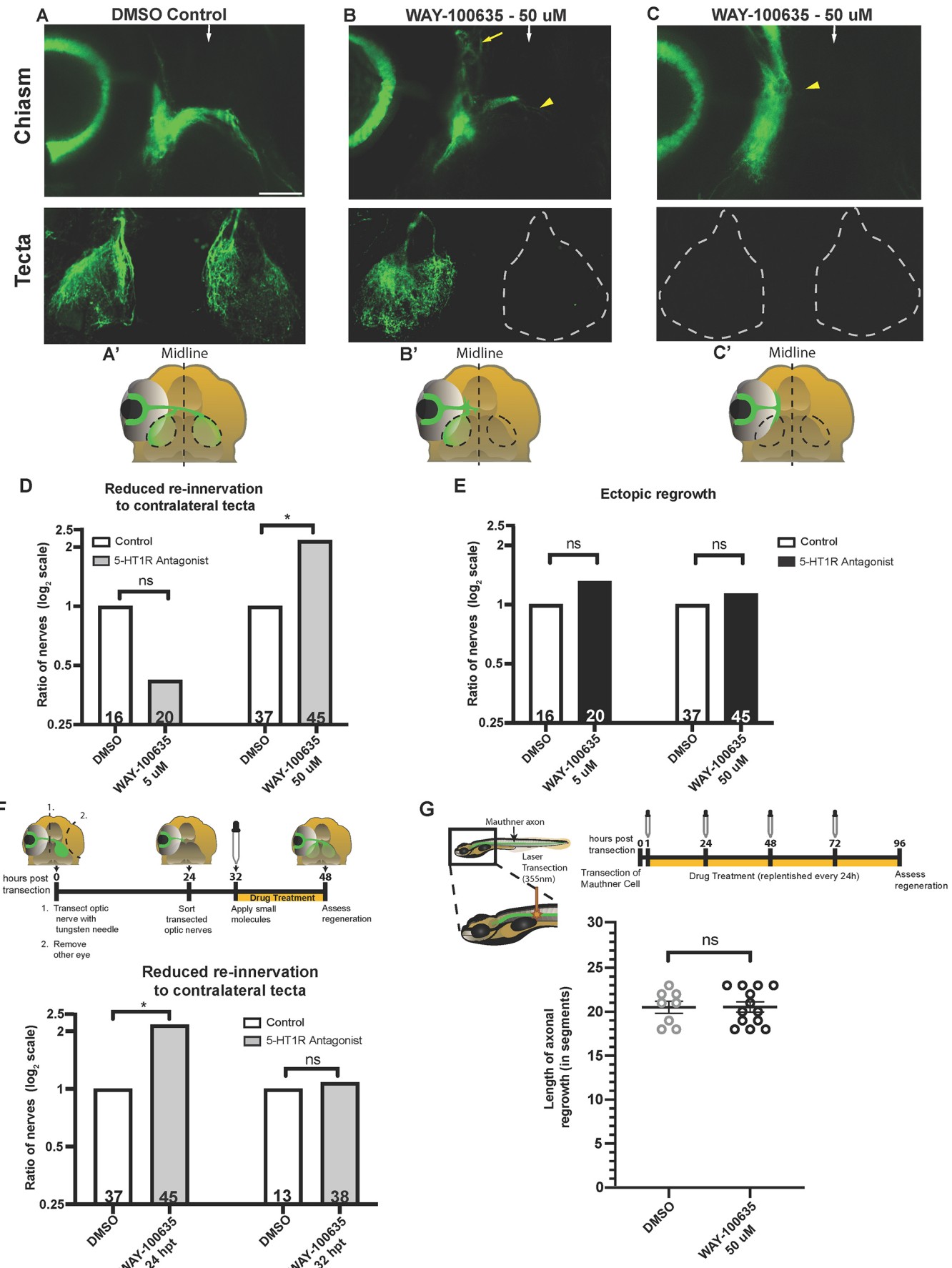

**Fig. 2.** See next page for legend.

**Fig. 2. 5-HT1 receptor signaling promotes early optic nerve regrowth without impairing overall CNS axonal regrowth.** (A-C′) Fluorescent representative images (A-C) and schematics (A′-C′) of *Tg(Isl2b:GFP)* at 48 hpt of larvae exposed to 0.3% DMSO (A) and 50 µM of antagonist WAY-100635 (B,C). Images in B and C are examples of two different types of regrowth observed when applying this small molecule. White arrows indicate the midline region at the optic chiasm. Scale bar: 50 µm. (B) RGC axons of larvae treated with the antagonist WAY-100635 regrew toward the optic chiasm (top, yellow arrowhead) but failed to innervate the contralateral tectum (dashed lines, bottom). A group of axons regrew ectopically away from the optic chiasm (yellow arrow). Re-innervation of the ipsilateral tectum begins soon after the small molecules are added. (C) Regenerating RGC axons of a larva treated with the antagonist WAY-100635 stalled near the injury site (yellow arrowhead, top). No axonal regrowth is observed in the optic chiasm or contralateral tectum (dashed lines, bottom). (D) Quantification of optic nerve axonal re-innervation to contralateral tectum at 48 hpt in *Tg(Isl2b:GFP)* larvae treated from 24 hpt to 48 hpt with 0.3% DMSO or WAY-100635. Bars represent the relative rate of optic nerves that fail to re-innervate the contralateral tectum in WAY-100635-treated (gray bars) and control (white bars) groups, plotted as a $\log_2$ scale. See Materials and Methods for details on ratios and fold-change calculations. *$P<0.05$ (two-tailed Fisher's exact test). ns, not significant. $n=21$ and $n=25$ for nerves treated with DMSO and 5 µM WAY-100635, respectively; $n=37$ and $n=45$ for nerves treated with DMSO and 50 µM WAY-100635, respectively. (E) Quantification of ectopic regrowth at 48 hpt in *Tg(Isl2b:GFP)* larvae treated from 24 hpt to 48 hpt with 0.3% DMSO or WAY-100635. Bars represent the relative rate of optic nerves' ectopic regrowth in WAY-100635-treated (black bars) and control (white bars) groups, plotted as a $\log_2$ scale. See Materials and Methods for details on ratios and fold-change calculations. The data displayed in D,E were obtained from larvae in the same treated groups. Statistics were determined using the two-tailed Fisher's exact test; ns, not significant. $n=21$ and $n=25$ for nerves treated with DMSO and 5 µM WAY-100635, respectively; $n=37$ and $n=45$ for nerves treated with DMSO and 50 µM WAY-100635, respectively. (F) Top: Timeline of optic nerve regeneration from 0 hpt-48 hpt. DMSO or small molecules were added to *Tg(Isl2b:GFP)* larvae at 32 hpt, and optic nerve regeneration was assessed at 48 hpt. Larvae were kept in 1× PTU/E3 from 24 hpt to 32 hpt. Bottom: Quantification of optic nerve axonal re-innervation to contralateral tecta at 48 hpt in *Tg(Isl2b:GFP)* larvae treated with 0.3% DMSO (white bars) or 50 µM WAY-100635 at 24 and 32 hpt (gray bars). Bar graphs and the ratios observed were calculated as detailed in D and Materials and Methods. The data showing 50 µM WAY-100635 at 24 hpt is identical to that shown in D and is shown here for visual comparison only. *$P<0.05$ (two-tailed Fisher's exact test); ns, not significant. $n=37$ and $n=45$, for nerves treated with DMSO and WAY-100635 at 24 hpt, respectively. $n=13$ and $n=38$, for nerves treated with DMSO and WAY-100635 at 32 hpt, respectively. (G) Top: Timeline of Mauthner axon regeneration. Schematic shows a laser transection (orange laser) performed in *Tg(Tol-056:GFP)* 5 dpf larvae using a UV laser (355-nm wavelength) at the ninth spinal cord hemisegment. After transection, 0.3% DMSO or 50 µM WAY-100635 was added to larvae starting at 1 hpt (dropper). Fresh DMSO and small molecules were replenished at 24, 48 and 72 hpt. Mauthner axon regrowth was assessed at 96 hpt. Bottom: Quantification of axonal length regrowth at 96 hpt, measured in spinal cord segments in control and WAY-100635 treated groups. Statistics were determined using the one-way ANOVA test. ns, not significant. Error bars represent s.e.m.

RGC axon regeneration early during the regeneration process when RGC axons navigate towards the optic chiasm. In summary, our findings support the idea that 5-HT receptor-dependent neuromodulation plays an instructive and tightly regulated role during the early regrowth of RGC axons towards the optic chiasm.

## DISCUSSION

Here, we used a recently established optic nerve transection assay combined with the ease of small molecule treatment of larval zebrafish to uncover pathways that promote optic nerve regeneration. From a small molecule screen, we identified

four small molecules that impaired optic nerve regeneration by selectively targeting 5-HT1 receptor signaling or other components of the serotonin signaling pathway. Previous studies in adult teleost fish have shown that after spinal cord injury serotonin promotes motor neuron regeneration (Barreiro-Iglesias et al., 2015), as well as facilitating axonal regeneration of local spinal interneurons (Huang et al., 2021). To our knowledge, the findings reported here are the first to define an acute, *in vivo* role for serotonin receptor signaling in CNS axon and optic nerve regeneration.

Specifically, we find that 5-HT1 receptors promote optic nerve regrowth during the early stages of regeneration after regenerating RGC axons emerge from the nerve stump and begin to extend towards the optic chiasm. Moreover, 5-HT1 receptor signaling is likely dispensable during optic nerve development and in a separate CNS regeneration process, suggesting that 5-HT receptors play a selective role in the modulation of optic nerve regeneration. Finally, we show that ectopic activation of 5-HT1 receptors causes RGC axons to regrow ectopically, suggesting that serotonin receptor-dependent neuromodulation directs regenerating RGC axons towards the optic tectum.

## 5-HT1 receptor signaling promotes RGC axon regeneration but is likely dispensable during RGC axon development

An ongoing discussion in the field is whether individual molecules or even entire pathways promoting regeneration simply recapitulate developmental mechanisms. Over the past years, examples of genes and signaling pathways playing distinct roles in axonal growth during CNS regeneration compared to development have emerged (Becker and Becker, 2007; Kusik et al., 2010; Wyatt et al., 2010). Gene expression analyses performed after optic nerve injury in adult zebrafish reveal that, in addition to the expected increase in the expression of RGC intrinsic genes linked to development, growth-promoting genes not associated with this developmental process were also upregulated (Veldman et al., 2007; Saul et al., 2010).

Our *in situ* hybridization data reveal that 5-HT1 *htr1aa* and *htr1ab* receptors are expressed in RGCs during the early stages of optic nerve regeneration. Similarly, previous work has demonstrated that the same 5-HT1 receptors are expressed in the retina and optic tectum during the development of the optic nerve (Pei et al., 2016). Given the expression profile of these receptors, we inhibited 5-HT1 receptors during the specific time window when RGC axons began extending towards the chiasm either during regeneration or development. Our findings that 5-HT1 receptor signaling functions during early optic nerve regrowth rather than development support the hypothesis that optic nerve regeneration is not merely a recapitulation of developmental programs.

It is therefore tempting to speculate that this differential requirement simply reflects a higher level of genetic redundancy among 5-HT receptors during development compared to regeneration. Consistent with this interpretation, *in vivo* knockout studies of different 5-HT receptor subtypes in vertebrates (Saudou et al., 1994; Heisler et al., 1998) have failed to detect significant brain defects or even mild phenotypes in axonal patterning during development (Trakhtenberg and Goldberg, 2012). Moreover, pathways required for retinal axonal growth and guidance in zebrafish, such as heparan sulfate proteoglycans, are known to regulate this process with a tremendous amount of redundancy through the use of compensatory mechanisms during development (Poulain and Yost, 2015). Another possibility is that a different set of 5-HT receptors might function during optic nerve development. In

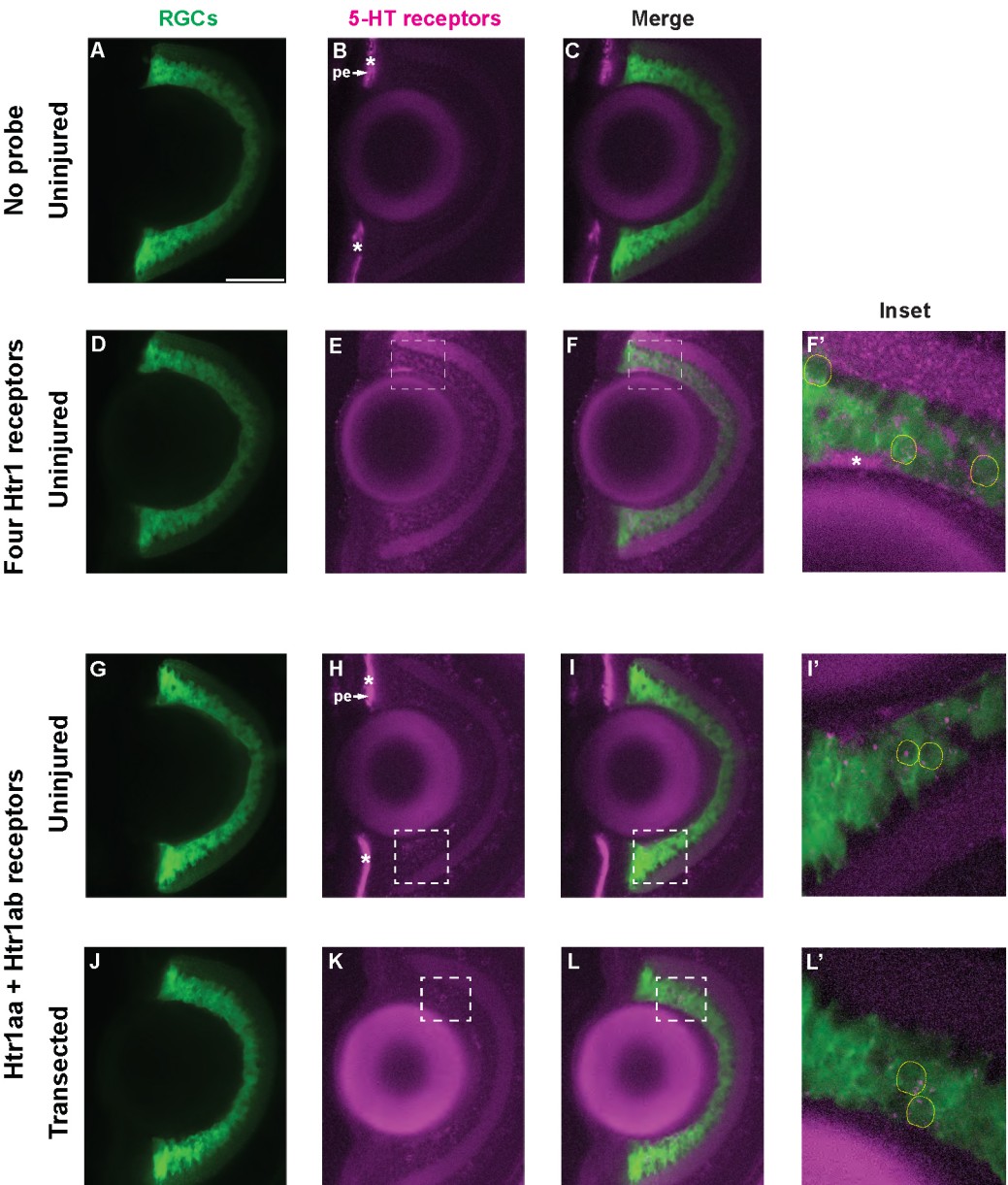

**Fig. 3. 5-HT1 receptor genes are expressed in RGCs at pre-transection and during optic nerve regeneration.** (A-F′) Representative images of retinas from *Tg(isl2b:GFP)* 5 dpf uninjured larvae stained with no probe control (A-C; *n*=8 retinas) or an *in situ* hybridization HCR probe mix for *htr1aa*, *htr1ab*, *htr1b* and *htr1d* (D-F′; *n*=12) (magenta). Images shown are maximum *z*-projections of six horizontal optical sections (32 μm). Dashed white boxes in E,F highlight the area enlarged F′, a merged maximum *z*-projection of ten horizontal optical sections (0.1 μm) showing multiple *Tg(isl2b:GFP)* RGC neurons (green, outlined with yellow dashed line) expressing mRNA of 5-HT1 receptors (magenta). In F′, the brightness of the green channel was adjusted for better visualization of 5-HT1 receptor genes inside RGC neurons. White asterisks depict nonspecific staining. The retinal pigmented epithelium (pe) was nonspecifically labeled by the amplifier with Alexa Fluor 546. (G-L′) Representative images of retinas from *Tg(isl2b:GFP)* 5 dpf uninjured larvae (G-I′; *n*=7) or larvae with transected optic nerves (J-L′; *n*=5) at 24 hpt stained with an *in situ* hybridization HCR probe mix for *htr1aa* and *htr1ab* (magenta). Images shown are maximum *z*-projections of seven horizontal optical sections (G-I) and 12 optical sections (J-L) (32 μm). Dashed white boxes in H,I highlight the area enlarged in I′, a merged maximum *z*-projection of 12 horizontal optical sections (0.1 μm). In I′, yellow dashed lines outline cell bodies with mRNA expression. The brightness of the green channel was adjusted for better visualization of 5-HT1 receptor genes inside RGC neurons. Dashed white boxes in K,L highlight the area enlarged in L′, a merged maximum *z*-projection of 11 horizontal optical sections (0.1 μm). In L′, yellow dashed lines outline cell bodies with mRNA expression. The brightness of the green channel was adjusted for better visualization of 5-HT1 receptor genes inside RGC neurons. White asterisks depict nonspecific staining. The retinal pigmented epithelium (pe) was nonspecifically labeled by the amplifier with Alexa Fluor 546. Scale bar: 50 μm (A-C,D-F,G-I,J-L).

fact, there are over 20 serotonin receptors in the zebrafish genome, most of which have their expression and function yet to be explored. Therefore, a thorough combinatorial analysis of 5-HT receptors will be crucial to determine whether 5-HT receptors play a redundant role in optic nerve development.

## 5-HT1 receptors selectively promote early regeneration of optic nerve axons toward the optic chiasm

Pioneering work on the regrowth of optic nerve axons after injury in spontaneous regeneration models has described the different stages of axon regeneration needed for the functional restoration of visual

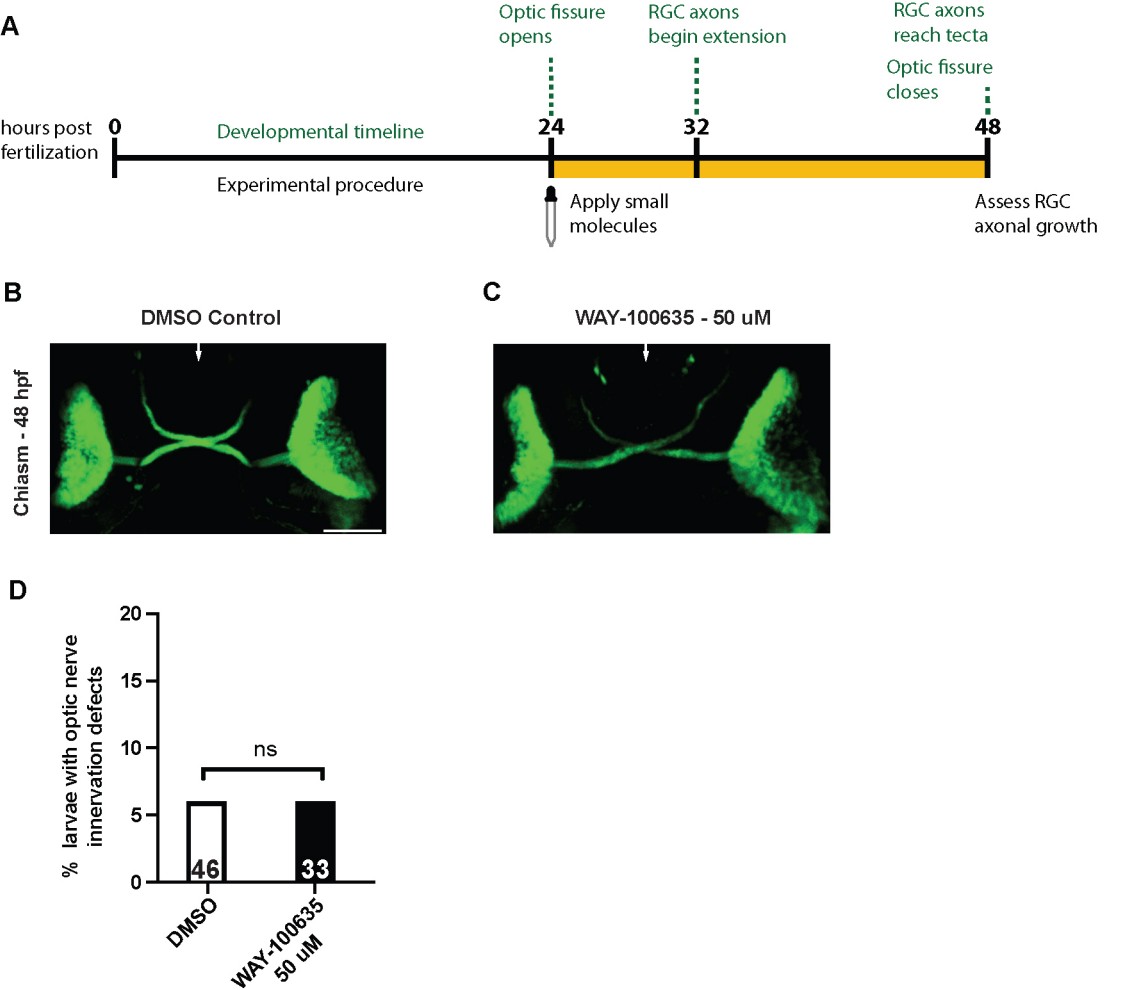

**Fig. 4. Optic nerve development is unaffected after treating zebrafish embryos with 5-HT1A antagonists.** (A) Timeline of optic nerve development from 24-48 hpf and larvae exposure to small molecules. DMSO (0.3%) or 50 µM of antagonist WAY-100635 was added to embryos at 24 hpf. Around 32 hpf, RGCs differentiate and immediately begin extending towards the brain. At 48 hpf, RGC axons have extended past the optic chiasm and have begun innervating the optic tecta. At this time point, treatment was washed off and optic nerve development was assessed. Developmental timeline information taken from Chhetri et al. (2014). (B,C) Representative images of 48 hpf *Tg(Isl2b:GFP)* embryos treated with 0.3% DMSO (B) and 50 µM WAY-100635 (C). White arrows depict the midline at the optic chiasm. No apparent growth or guidance defects are observed in the developing optic nerves of control and small molecule-treated larvae. Scale bar: 50 µm. (D) Quantification of optic nerve growth defects at 48 hpf in *Tg(Isl2b:GFP)* larvae treated from 24 hpf to 48 hpf with DMSO (white bar) or WAY-100635 (black bar). Bars represent the percentage of defective optic nerves for each treatment group. Statistics were determined using a two-tailed Fisher's exact test. ns, not significant. *n*=46 and *n*=33, for larvae treated with DMSO and WAY-100635, respectively.

projections (Bohn and Reier, 1985; Stuermer et al., 1992; Kaneda et al., 2008). These optic nerve regeneration stages first include that after injury RGC axons form a regenerative growth cone from the optic nerve stump; next, they extend toward the optic chiasm, across the chiasm; and, finally, they extend toward the optic tectum where they re-innervate the optic tectum (Becker and Becker, 2007; Diekmann et al., 2015b). These regeneration stages can be distinguished at the transcriptional level, as reflected by gene expression, including the upregulation of different transcription factors and pathways in RGCs during each stage (Dhara et al., 2019).

Our pharmacological inhibition of 5-HT1 receptors at distinct regeneration stages demonstrates that 5-HT1 receptors modulate the early extension of optic nerve axons towards the chiasm but are likely dispensable after crossing the chiasm. Thus, our findings that serotonin receptor signaling is selectively required for a particular regeneration stage are significant because they suggest that, similar to development, different genetic programs might control different stages of the optic nerve regeneration trajectory. For example, the

mTOR pathway is another signaling pathway that might regulate early optic nerve regeneration in larval zebrafish. In mammals, mTOR signaling plays a crucial role in potentiating the extension of regenerating RGC axons toward the chiasm (Park et al., 2008), and in adult zebrafish it is sharply upregulated when regenerating RGC axons grow towards the chiasm. Moreover, blocking mTOR activity inhibits optic nerve regeneration in adult fish (Diekmann et al., 2015a), a similar phenotype to the one we observe after blocking 5-HT receptor signaling. 5-HT receptors have previously been shown to interact with components of the mTOR pathway to regulate other CNS processes (Meffre et al., 2012; Teng et al., 2019), raising the possibility that these pathways might act together to promote the early stages of optic nerve regeneration.

### Serotonin receptor-dependent neuromodulation plays an instructive role in optic nerve regeneration

Our results demonstrate that inhibition of 5-HT1 receptors leads to reduced regeneration of optic nerve axons, while ectopic activation of

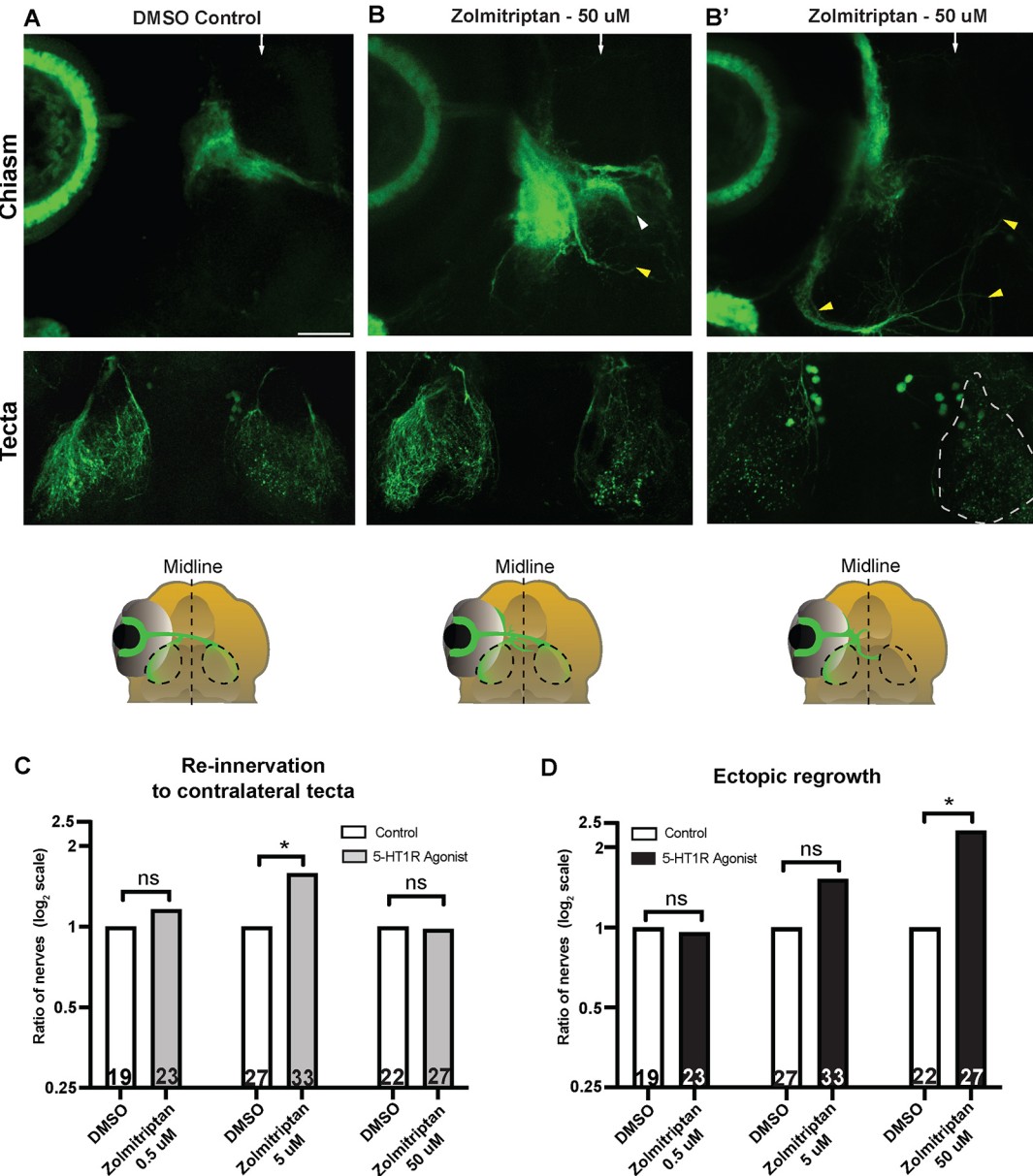

**Fig. 5. Exogenous activation of 5-HT1 receptors significantly increases axonal regrowth in a dose-dependent manner.** (A-B′) Fluorescent representative images and schematics of *Tg(Isl2b:GFP)* at 48 hpt of larvae exposed to 0.3% DMSO (A) and 50 µM of the agonist zolmitriptan (B,B′). Images in B and B′ are examples of two different types of regrowth observed when applying this small molecule. Scale bar: 50 µm. (B) Regenerating optic nerve of a larva treated with 50 µM zolmitriptan shows axons exhibiting ectopic regrowth at the optic chiasm (white arrowhead). RGC axons also regrew misguided away from the chiasm to a more posterior region of the brain (yellow arrowhead). Other RGC axons re-innervated both optic tecta by 48 hpt (bottom image). (B′) The regenerating optic nerve of a larva treated with 50 µM zolmitriptan shows a significant amount of axon fascicles ectopically regrowing from the injury site and extending away from the optic chiasm (yellow arrowheads). RGC axons did not re-innervate the contralateral tectum by the 48 hpt time point (bottom image, dashed lines). White arrows depict the midline at the optic chiasm. (C) Quantification of optic nerve axonal re-innervation to the contralateral tectum at 48 hpt in larvae treated from 24 hpt to 48 hpt with 0.3% DMSO or the agonist zolmitriptan. Bars represent the relative rate of optic nerves that re-innervated the contralateral tectum in zolmitriptan-treated (gray bars) and control (white bars) groups, plotted as a log₂ scale. See Materials and Methods for details on ratios and fold-change calculations. The data displayed in C,D were obtained from larvae in the same treated groups. *$P<0.05$ (two-tailed Fisher's exact test); ns, not significant. $n=19$ and $n=23$ for nerves treated with DMSO and 0.5 µM zolmitriptan, respectively; $n=27$ and $n=33$ for nerves treated with DMSO and 5 µM zolmitriptan, respectively; and $n=22$ and $n=27$ for nerves treated with DMSO and 50 µM Zolmitriptan, respectively. (D) Quantification of ectopic regrowth at 48 hpt in *Tg(Isl2b:GFP)* larvae treated from 24 hpt to 48 hpt with 0.3% DMSO or zolmitriptan. Bars represent the relative rate of optic nerves' ectopic regrowth in zolmitriptan-treated (black bars) and control (white bars) groups, plotted as a log₂ scale. See Materials and Methods for details on ratios and fold-change calculations. *$P<0.05$ (two-tailed Fisher's exact test); ns, not significant. $n=19$ and $n=23$ for nerves treated with DMSO and 0.5 µM zolmitriptan, respectively; $n=27$ and $n=33$ for nerves treated with DMSO and 5 µM zolmitriptan, respectively; and $n=22$ and $n=27$ for nerves treated with DMSO and 50 µM zolmitriptan, respectively.

5-HT1 receptors leads to a partially opposite phenotype, resulting in increased regeneration and, at higher agonist levels, to misguided regrowth. This suggests that 5-HT1 receptors are crucial for optic nerve regeneration and might, in fact, instruct the regeneration of RGC axons as they emerge from the nerve stump and navigate toward the optic chiasm. How might serotonin receptor-dependent signaling instruct regenerating axons during optic nerve regeneration? Recent work has shown that serotonin acts as a guidance cue when presented

to growth cones in neuronal cultures. Specifically, growth cone repulsion is regulated through the 5-HT1B receptor and its inhibition of cyclic AMP (cAMP) levels (Vicenzi et al., 2021). Given the expression of different 5-HT1 receptor genes in RGC neurons during regeneration (this study), a potential mechanism is that serotonin could act as a guidance cue along the optic nerve tracts and that its binding to 5-HT1 receptors on RGC axons repulses axons away from the retina and towards the optic chiasm. These receptors may, in turn, modulate cAMP signaling, which is known to be pivotal for promoting RGC axon regeneration in different models (Rodger et al., 2005; Hellström and Harvey, 2014).

### Conclusions and limitations
While pharmacological approaches are a strength of our study, one limitation is that we cannot entirely exclude the possibility that these approaches may lead to some off-target effects *in vivo*. Moreover, due to the potential pleiotropic effects of 5-HT modulation in the CNS, 5-HT receptor signaling might indirectly promote optic nerve regrowth to the CNS midline through neuromodulation of yet-to-be-defined guidance cues. This interpretation is supported by previous research elucidating the role of 5-HT1B/1D receptor signaling in mediating the response of thalamocortical axons to netrin (Bonnin et al., 2007) and 5-HT2 receptors directing the midline crossing of commissural axons by regulating the translation of ephrin B2 (Xing et al., 2015). Therefore, using conditional knockouts and similar molecular methods to target specific 5-HT receptors is an important next step to further our understanding of 5-HT receptor signaling in CNS and optic nerve regeneration.

Our findings are consistent with a model in which tight regulation of 5-HT1 receptor signaling levels is needed to effectively direct regenerating axons toward the chiasm. In this model, a dial indicator can be used to envision the 5-HT1 signaling levels that lead to a particular optic nerve regeneration phenotype. A dial pointing to normal 5-HT1 receptor signaling levels would correlate with the levels required for axons to extend and maintain their pre-transection trajectory after injury. Decreased levels of 5-HT1 receptor signaling, the levels likely attained after blocking 5-HT1 receptors with the antagonist WAY-100635, impede optic nerve regrowth. Conversely, an increase in 5-HT1 receptor signaling, similar to the levels attained after ectopic activation with 5 μM of the agonist zolmitriptan, leads to enhanced optic nerve regrowth, culminating in higher rates of innervation to the contralateral tectum. Finally, increasing 5-HT1 receptor signaling levels further with 50 μM of zolmitriptan causes significant ectopic optic nerve regrowth, possibly due to a 5-HT1 receptor-mediated increase in the repulsion of axons in response to serotonin. Future work is required to determine whether 5-HT receptor signaling promotes early optic nerve regeneration after injury in mammalian systems.

## MATERIALS AND METHODS
### Fish maintenance and ethics statement
All animal protocols were approved by the University of Pennsylvania Institutional Animal Care and Use Committee (IACUC). *Danio rerio* transgenic lines were maintained in the Tübigen or Tupfel long fin (TLF) genetic background and raised as described by Mullins et al. (1994). *Tg(isl2b:GFP)* (Pittman et al., 2008) larvae were maintained in dishes with phenylthiourea (1× PTU, 0.2 mM in E3 medium) beginning at 4 hpf and incubated in the dark at 29°C to decrease melanocyte pigmentation.

### Transection assay and small molecule screen
At 5 dpf, larvae were anesthetized using 0.0053% tricaine in a PTU/E3 solution and then mounted on a glass microscopy slide using 2.5% low-melt

agarose (SeaPlaque, Lonza) in a PTU/E3 solution containing 0.016% tricaine. Transections were performed on an Olympus SZX16 fluorescent microscope, as detailed by Harvey et al. (2019), with some modifications. Briefly, the optic nerve of the left eye was transected using a sharpened tungsten needle (Fine Science Tools), and the right eye was removed using sharpened forceps. Fish were detached from the agar and allowed to recover in dishes containing 1× Ringer's Solution for 1 h.

Complete optic nerve transection was confirmed by examining transected larvae at 22-24 hpt. Larvae that displayed no axonal GFP remnants from the injury site to the tectum were added to 48-well plates and treated with 0.3% DMSO or small molecules from 24 hpt to 48 hpt unless otherwise noted. Small molecules used in the screen were obtained from the University of Pennsylvania High-Throughput Screening Core (Selleckchem Bioactive: FDA-approved/FDA-like small molecules) (Lamire et al., 2023). The known targets of small molecules in the drug library were classified based on their genetic pathways using the PANTHER Classification System (pantherdb.org) (Mi and Thomas, 2009). To determine the number of genetic pathways that form part of the drug library, we submitted a target list provided by Selleckchem to the 'Gene List Analysis' section in PANTHER. Targets not recognized by the program were manually examined using the PANTHER 'Prowler' Pathway section. Stock and working small molecule solutions were prepared as described previously (Lamire et al., 2023). Briefly, stock solutions (100× frozen stocks in DMSO) were initially diluted 1:100 in E3, to obtain a 10× solution. Then, 30 μl of this solution was added into the wells, yielding a 10 μM drug concentration in 0.3% DMSO. Three unique small molecules were added to 'small molecule pools' to determine their effect on optic nerve regeneration. Usually, each drug pool was tested on six larvae as part of an experimental group. If the result was unclear, the test, including the re-testing of DMSO-treated control groups, was repeated. Pools that caused impaired regeneration in more than half of the tested larvae (usually four out of six) were called 'hits'. Investigators were unaware of the identity of the small molecules in each drug pool. The experimental and control groups were treated in the same order as optic nerve regeneration was assessed to ensure close to equal treatment duration. Small molecule treatments that resulted in lethality, altered body morphology, or a significant reduction in responsiveness to touch in larval zebrafish during the treatment period were excluded from further testing. All small molecules in drug pools that impaired regeneration were then tested individually to validate the phenotypes observed using the assay described above. The identity of pools and compounds was revealed only after scoring regeneration, reducing the subjective bias. Usually, each individual small molecule was tested on eight larvae as part of an experimental group. Pools that caused impaired regeneration in more than half of the tested larvae (usually six out of eight) were validated and analyzed further, including in dose-response and time-course experiments, many of which were highlighted in this publication. Individual small molecules that impaired regeneration were re-tested using a different batch of the same small molecule, confirming they impaired optic nerve regeneration. All zebrafish larvae were randomly distributed into a control and experimental group for each experimental batch.

### Individual small molecule treatment
For individual treatment with small molecules, a 2.5 mM stock of WAY-100635 (Sigma-Aldrich, W108) was prepared by dissolving a new vial of 10 mg of WAY-100635 powder in 7.42 ml of 100% DMSO. The stock solution was then further dissolved in PTU/E3 to a final concentration of 5 μM or 50 μM (0.3% DMSO final concentration). The stock solution of WAY-100635 was freeze-thawed a maximum of two times and then disposed of. A 50 mM stock of zolmitriptan (Sigma-Aldrich, SML0248) was prepared by dissolving a new vial of 10 mg of zolmitriptan powder in 696 μl of 100% DMSO. The stock solution was then further dissolved in PTU/E3 to a final concentration of 0.5 μM, 5 μM or 50 μM (0.3% DMSO final concentration). The stock solution of zolmitriptan was freeze-thawed a maximum of two times and then disposed of. Compounds were applied to wells in a 48-well plate dish containing fully transected 6 dpf larvae at different time points, as indicated in the Results section. Control larvae received a solution of 0.3% DMSO in PTU/E3. Isl2b:GFP+ larvae were pooled and randomly assigned to control and experimental groups for each

individual replicate performed. Each control pool contained at least five samples (i.e. five optic nerves) that were analyzed during the optic nerve regeneration assay.

## Quantification of optic nerve regrowth phenotypes

For quantification of optic nerve re-innervation to the contralateral tectum following optic nerve transection, a defect was scored when a regenerating optic nerve (including any fascicle or individual axon part of this nerve) failed to re-innervate the optic tectum at 48 hpt. For quantification of optic nerves showing a misguided growth defect, an ectopic regrowth phenotype was scored when axons regrew at a 30° angle or more away from the stereotypical trajectory that regenerating axons follow toward the optic chiasm in pre-transected animals. Results are displayed in bar graphs representing the ratio of optic nerves that displayed a defect in drug-treated or control groups. Ratios were calculated to normalize the results obtained from each experiment. The ratio for control groups (white bars) is always 1. The ratios for drug-treated groups were calculated as follows:

$$\frac{\# \text{ of [drug treated]ON with defect/total \# of [drug treated]ON analyzed}}{\# \text{ of [control treated]ON with defect/total \# of [control treated]ON analyzed}},$$

where ON stands for optic nerve.

Quantification of Mauthner cell axon regrowth at 96 hpt was performed as described by Bremer et al. (2019). The length of individual Mauthner axons was measured in spinal cord hemisegments and compared between the control group (0.3% DMSO treated) and drug-treated group. See Table S2 for the raw data obtained during the quantification of optic nerve regrowth and Mauthner cell axon regrowth.

## Statistical analysis

For statistical calculations of optic nerve regeneration phenotypes, we compared control and experimental groups on a contingency table. We applied a two-tailed Fisher's exact test for categorical outcomes between two samples using the online calculator provided at http://www.quantitativeskills.com/sisa/statistics/fisher.htm (retrieved 25 April, 2023). For statistical analysis related to Mauthner axonal regeneration, we applied the one-way ANOVA test. Computations were performed using the Prism 10 software package from GraphPad. Graphs were generated using Prism 10 (GraphPad).

## Immunostaining and confocal imaging

Zebrafish larvae were stained as previously described (Harvey et al., 2023). Larvae were incubated with the primary antibody mouse anti-GFP (1:200; JL-8, BD Biosciences) and secondary antibody goat anti-mouse Alexa 488 (1:500; A32723, Thermo Fisher Scientific) overnight at 4°C. Antibodies were diluted in 1% bovine serum albumin and 1% DMSO in PBS + 0.1% Tween 20. Stained larvae were mounted in Vectashield (Vector Laboratories) for confocal imaging. The brains of zebrafish larvae were imaged on a Zeiss LSM 880 confocal microscope using the 20× and 40× objectives. Image stacks of optic chasms or optic tecta were compressed into maximum-intensity projections. All images shown were adjusted for brightness and contrast, and color was assigned using Fiji.

## FISH with HCR

*htr1aa*, *htr1ab*, *htr1b* and *htr1d* mRNA expression were detected by FISH with HCR (Molecular Instruments) (Choi et al., 2018). HCR probes, buffers and hairpins were purchased from Molecular Instruments. *Tg(isl2b:GFP)* larvae were fixed at 5 dpf with 4% paraformaldehyde in PBS overnight at 4°C in a rotating wheel. The staining was performed as previously described (Shainer et al., 2023) using B1 and B2 amplifiers with Alexa Fluor 546.

## Acknowledgements

We thank Andrea Stout of the University of Pennsylvania CDB Microscopy Core for providing technical assistance with our confocal imaging and Granato lab members for their comments and discussions.

## Competing interests

The authors declare no competing or financial interests.

## Author contributions

Conceptualization: K.S.-S., M.G.; Data curation: K.S.-S., M.B.; Formal analysis: K.S.-S., M.G.; Funding acquisition: M.G.; Investigation: K.S.-S., M.B., J.M.; Methodology: K.S.-S., M.G.; Project administration: M.G.; Resources: M.G.; Supervision: M.G.; Validation: K.S.-S.; Visualization: K.S.-S., J.M., M.G.; Writing – original draft: K.S.-S.; Writing – review & editing: K.S.-S., M.G.

## Funding

This work was supported by grants from the National Institutes of Health (R01EY024861 to M.G. and K12GM081259 to K.S.-S.). Open Access funding provided by the University of Pennsylvania. Deposited in PMC for immediate release.

## Data and resource availability

All relevant data and details of resources can be found within the article and its supplementary information.

## Peer review history

The peer review history is available online at https://journals.biologists.com/dev/lookup/doi/10.1242/dev.204334.reviewer-comments.pdf

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
