## [Peer Review File · Development (Cambridge, England)]

Serotonin neuromodulation directs optic nerve regeneration

Kristian Saied-Santiago, Melissa Baxter, Jaffna Mathiapparanam and Michael Granato

DOI: 10.1242/dev.204334

Editor: Steve Wilson

Review timeline

Original submission:	19 August 2024
Editorial decision:	21 October 2024
First revision received:	18 February 2025
Editorial decision:	8 April 2025
Second revision received:	28 May 2025
Accepted:	29 May 2025

Original submission

First decision letter

MS ID#: dev.204334

MS TITLE: Serotonin neuromodulation directs optic nerve regeneration

AUTHORS: Kristian Saied-Santiago; Melissa Baxter; Jaffna Mathiapparanam; Michael Granato

Dear Michael,

I have now received all the referees' reports on the above manuscript, and have reached a decision. The referees' comments are appended below, or you can access them online: please go to:

As you will see, the referees differ in their enthusiasm for publishing your work in Development and all have suggestions for improvements. Referee 2 is the most critical and suggests additional genetic and transgenic manipulations to strengthen the work. While such studies would no doubt strengthen the study, they may be beyond the scope of what you are willing to do for the revision and I would understand if you do not fully address all suggestions raised. There are many other suggestions from this and the other referees that I think can be addressed and should improve the study. If you are able to revise the manuscript along the lines suggested, I will be happy receive a revised version of the manuscript.

Please also note that Development will normally permit only one round of major revision. If it would be helpful, you are welcome to contact us to discuss your revision in greater detail. Please send us a point-by-point response indicating your plans for addressing the referees' comments, and we will look over this and provide further guidance.

Please attend to all of the reviewers' comments and ensure that you clearly highlight all changes made in the revised manuscript. Please avoid using 'Tracked changes' in Word files as these are lost in PDF conversion. I should be grateful if you would also provide a point-by-point response detailing how you have dealt with the points raised by the reviewers in the 'Response to Reviewers' box. If you do not agree with any of their criticisms or suggestions please explain clearly why this is so.

Reviewer 1*Advance summary and potential significance to field*

This manuscript reports a previously uncharacterized function of serotonin type-1 receptors (5-HT1) in optic nerve (ON) regeneration in vivo. Using an elegant ON transection assay in zebrafish combined with a small molecule library screen, the authors identified several compounds that modulate serotonin signaling and modified ON regeneration. The authors then demonstrate that 5-HT1 receptors are expressed by retinal ganglion cells (RGCs) before transection and during ON regeneration, and that inhibiting 5-HT1A receptors reduces regeneration but does not affect developmental axon outgrowth. The authors finally show that activating 5-HT1B/D receptors can enhance regeneration. Overall, this study makes a significant contribution to the field by uncovering a novel and specific function of 5-HT1 receptors in ON regeneration.

Comments for the author

The study is well carried out, technically of high standard, and provides a thoughtful discussion about the functions of serotonin signaling in the regulation of nerve regeneration. It is appropriate for publication in *Development* provided the authors address the concerns detailed below:

- 1) One major concern is the use of a single 5-HT1A antagonist (WAY-100635) and a single 5-HT1B/1D agonist (zolmitriptan) to define the role of 5-HT1 receptors in ON regeneration. Different dose-dependent responses were observed for each treatment, which might be explained by different affinities of the drugs for their targets. For instance in figure 2D, 5 μ M of WAY-100635 seems to cause a decrease (although not significant) in ratio of nerves that contrasts with the strong increase observed with 50 μ M. Since WAY-100635 also has a high affinity for D4 receptors, its effects should be confirmed with a second, independent 5-HT1A antagonist or by analyzing *htr1aa/ab* crispants. Similarly, the functions of 5-HT1A, 5-HT1B and 5-HT1D should be further examined using a 5-HT1A agonist and 5HT1B/1D antagonists.
- 2) Figure 2F: the authors applied WAY-100635 at 32 hpf to assess the effects of 5-HT1A inhibition after axons have reached the optic chiasm. Where are axons at 32 hpt (or just after 32 hpt when the antagonist is active)? Have they crossed the chiasm and already entered the optic tract, or have they just reached the chiasm but not crossed it yet? It would be interesting to clearly pinpoint when, during their navigation, RGC axons lose their sensitivity to the antagonist. Adding a picture of RGC axons at 32 hpt might clarify this point.
- 3) Figure 3 shows a clear expression of *htr1aa* and *htr1ab* in RGCs in the uninjured retina and at 24 hpt. Are *htr1aa* and *htr1ab* still expressed after 32 hpt, when application of WAY-100635 does not have any effect on regeneration?
- 4) The authors provide a nice discussion on the possible mechanisms whereby 5-HT1 receptors regulate ON regeneration. Is serotonin observed along ON tracts after transection? Can 5-HT1 receptors be detected along regenerating axons (it might be difficult to answer this last question considering the paucity of antibodies working in zebrafish)?
- 5) The authors indicate in Table 1 that applying serotonin reduced tectal re-innervation, which contrasts with the effects of zolmitriptan and WAY-100635. Are there other serotonin receptors expressed in the visual system that could explain this result?

Minor points:

> In material and methods, please clarify the formula used to calculate ratios for drug treated groups: what happens to the ratio if the # of [control treated] ON nerves with defect is zero (does this happens)? Also, isn't "ON nerves" redundant?

- > Figure 3: please indicate whether optical sections are coronal or longitudinal.
- > In the text describing Figure 4, the conclusion should be that 5-HT_{1A} (not 5-HT₁) receptor function is dispensable for RGC axon outgrowth. Since WAY-100635 is specifically targeting 5-HT_{1A} receptors, a role for 5-HT_{1B} or 5-HT_{1D} cannot be ruled out.
- > Panels A-F' in Figure S2 should be combined with Figure 3 (I wondered what the expression of *htr1b* and *htr1d* was before finding out the answer later in the paper). Panels G and H could then be added to Figure 5.
- > Figure 5: Showing pictures illustrating the effects of both concentrations of zolmitriptan would strengthen the study. Also, the manuscript indicates that panels A-D correspond to 5 μ M of zolmitriptan, but the legend indicates 50 μ M.

Reviewer 2

Comments for the author

The authors investigate the role of serotonin (5-HT) signaling in optic nerve regeneration using larval zebrafish as a model. They conducted a small molecule screen and identified serotonin type-1 (5-HT₁) receptor modulators that impact optic nerve regeneration. Their findings suggest that inhibiting 5-HT₁ receptors or components of the 5-HT pathway selectively impedes regeneration, while agonist-mediated activation enhances axonal regrowth. The authors conclude that 5-HT₁ receptors play a critical role in promoting early stages of optic nerve regeneration and propose that serotonin-dependent neuromodulation directs optic nerve regeneration in vivo. The manuscript provides preliminary insights into the role of serotonin signaling in optic nerve regeneration in zebrafish. However, the study has several critical flaws and would benefit from more evidence to support the claims. To be suitable for publication, the paper requires extensive revisions and substantial additional work.

I am particularly concerned about the following points:

- The manuscript relies heavily on pharmacological inhibition and activation of serotonin receptors using small molecules. While this approach is valuable for initial screening, it is insufficient to establish a direct, specific role for serotonin receptors in optic nerve regeneration. The use of complementary genetic approaches, such as CRISPR/Cas9-mediated knockouts, RNA interference, or transgenic overexpression of specific serotonin receptor subtypes (e.g., 5-HT_{1A}, 5-HT_{1B}) in zebrafish, would provide more direct evidence for the role of these receptors. For example, the manuscript claims that 5-HT₁ receptor signaling is critical for axonal regrowth but fails to distinguish whether this effect is due to receptor activity in RGCs or other cell types. Genetic tools could allow for cell type-specific manipulation, thereby strengthening the claims of receptor involvement in RGC axonal regeneration.
- The manuscript describes a small molecule screen using a library of FDA-approved compounds to identify modulators of serotonin signaling. However, the criteria for selecting the serotonin-related targets are not sufficiently detailed. The authors could clarify how compounds were chosen for follow-up studies, including their specificity, known off-target effects, and pharmacokinetic properties. For example, why were only the four serotonin-targeting compounds selected for detailed study, and what evidence supports their specificity for serotonin receptors in the context of zebrafish? Additionally, details on the validation of hits from the small molecule screen are sparse. The authors could provide a more comprehensive description of the follow-up experiments conducted to confirm the initial screen findings, such as dose-response tests, time-course experiments, and verification in independent replicates.
- The authors may consider addressing potential sources of bias or error in the experimental design. For instance, while the small molecule treatments were blinded, it is unclear if the subsequent imaging and quantification were also conducted blind to treatment conditions. Unblinded assessment of regeneration could introduce bias in quantifying regrowth or misguidance. The authors could provide more information on whether all stages of the experiment, including the assessment of axon regrowth and statistical analysis, were blinded and how randomization was achieved. Additionally, there is no

discussion of the reproducibility of the findings across different experimental batches, which is critical to ensure the reliability of the reported effects.

- While the manuscript presents data on the effects of serotonin signaling modulation on optic nerve regeneration, it does not provide functional validation of the proposed mechanisms. For example, the study suggests that 5-HT1 receptors direct regenerating axons toward the optic chiasm by modulating growth cone guidance, possibly via cAMP signaling pathways. However, no direct evidence, such as measurements of intracellular cAMP levels, or experiments manipulating downstream signaling components, is provided to support this mechanism. The authors could consider performing functional assays to validate these claims, such as live-cell imaging of growth cones or molecular assays to quantify changes in second messengers (e.g., cAMP) upon receptor activation or inhibition. This would provide more concrete evidence to support the proposed mechanistic pathways.
- The manuscript would benefit from a more comprehensive description of the control experiments used to validate the specificity of the findings. For example, it is critical to demonstrate that the observed effects on optic nerve regeneration are specifically due to the modulation of serotonin signaling and not secondary to general toxicity or off-target effects. The authors mention using a DMSO control but could also include additional controls, such as vehicle-only controls for each small molecule, and parallel experiments using structurally unrelated inhibitors of serotonin receptors to demonstrate that the effects are specifically mediated through serotonin signaling. Moreover, for the genetic manipulations suggested above, appropriate controls, such as scrambled or non-targeting controls, would be necessary to validate the specificity of the observed phenotypes.
- The manuscript describes the use of serotonin receptor agonists and antagonists at different concentrations and time points but requires a discussion on the temporal dynamics of serotonin signaling during optic nerve regeneration. The authors claim that serotonin signaling is particularly important during the early stages of regeneration, but the manuscript could include time-course studies to map out when serotonin receptor activity is most critical. To strengthen the findings, the authors could perform and discuss time-course experiments to identify the precise temporal windows when serotonin receptor modulation affects regeneration. This would provide a more detailed understanding of how serotonin signaling integrates into the broader context of regenerative processes.
- The manuscript's methods section could be improved with more and sufficient detail in several areas that are critical for reproducibility and interpretation of the findings. For example, there is insufficient information about how the small molecule concentrations were chosen, how often treatments were refreshed, or how long the treatments were applied relative to key developmental or regenerative milestones. The manuscript could provide a more detailed methodological description, including justifications for all chosen experimental conditions, to ensure that other researchers can replicate the findings.
- The authors use small molecule inhibitors and agonists, which may have off-target effects that could confound the interpretation of the results. For instance, the use of WAY-100635, a known 5-HT1A receptor antagonist, may also have affinity for other receptors or signaling pathways. The manuscript could discuss these potential off-target effects and consider performing additional experiments, such as using more selective inhibitors or genetic approaches, to confirm that the observed effects are truly due to serotonin receptor modulation.

Major comments:

- The manuscript would be stronger if it considered the relevant literature, especially studies related to serotonin receptor signaling in CNS regeneration, including brain regeneration in zebrafish and other models. For example, the claim that this study is the first to define an acute, in vivo role for serotonin receptors in CNS axon regeneration is not accurate. Several studies have already shown relevance of serotonergic signaling for CNS axon regeneration (PMID 34876587, 31905199, 26565906 and others). The authors must provide a more thorough and balanced discussion that includes studies supporting and contradicting their findings.
- The localization of serotonin receptors is not clearly demonstrated in the manuscript. The in situ hybridization is vague and combines probes, making it difficult to confirm the specific expression of these receptors in retinal ganglion cells (RGCs). The authors could utilize existing single-cell RNA sequencing (scSeq) datasets for zebrafish to externally validate their claims about receptor expression, as this is crucial to substantiate the specificity of their findings.
- It would be useful to consider the pleiotropic effects of serotonin inhibition. Given that the entire larvae are treated with antagonists or agonists, the specificity of the observed effects on optic nerve

regeneration is questionable. The authors must address the potential for broad, non-specific effects on multiple serotonin pathways and provide a detailed discussion of these limitations.

- The figures need essential details, such as individual data points and distribution, which prevents a clear understanding of data variance. The authors could revise the graphs to include individual data points and provide a more detailed description of the variance in their data. Additionally, the manuscript will benefit from a discussion on the appropriateness of the statistical methods used. For example, it is unclear if the sample sizes are sufficient for the conclusions drawn, and whether proper statistical controls were employed.
- The quantification approach for spinal cord regeneration in the manuscript is not well justified. The authors observe no change in regeneration but may be using an inappropriate readout. The authors could consider alternative readouts or regions to quantify regenerative changes, possibly by comparing with established models or metrics used in similar studies. Several studies showed changes in axon regeneration in spinal cord with serotonin modulation.
- The observed differences between high and low doses of antagonists suggest pleiotropic effects of the treatments. It would be beneficial if the manuscript provided a dose-response curve and discussed the implications of the pleiotropic effects of their treatments.
- The manuscript should contain a paragraph discussing the limitations of the study, which is crucial in this case. The authors could add a comprehensive limitations section to acknowledge these issues and provide context for the interpretation of their findings.

Given the preliminary nature of the findings and the substantial revisions needed, I recommend a major revision. The authors must address the issues raised above and provide additional experimental data to support their claims more convincingly.

- Authors could expand the literature review. They could include recent studies on serotonin receptor signaling in CNS regeneration, particularly those that both support and contradict the findings of this study. Acknowledging the existing work and discussing how this study adds to or differs from previous research would make sense.
- Authors could clarify the receptor localization. They can perform additional experiments or use available single-cell RNA sequencing datasets to validate the expression of serotonin receptors in the specific neuronal populations of interest. This will strengthen the evidence for receptor involvement in optic nerve regeneration.
- Discussing the drug specificity and pleiotropic effects would be useful. Authors can provide a more in-depth discussion of the potential pleiotropic effects of the serotonin receptor modulators used in this study.
- Including a section for limitations of this study will be highly beneficial.
- Authors could revise data presentation by updating the figures to show individual data points and clearly present the variance. If needed, they could reconsider the statistical considerations, such as reviewing and potentially revising the methods used for quantifying regeneration, especially in the spinal cord regeneration assays. Providing scientific reasoning for the choice of readout and consider alternative methods or areas for quantification will help the audience by reducing the guesswork.
- Performing a dose-response analysis will strengthen the conclusions. This can also help a discussion on how varying concentrations may lead to different effects and what this implies for the specificity of the observed phenomena.

Reviewer 3

Advance summary and potential significance to field

The manuscript "Serotonin neuromodulation directs optic nerve regeneration" provides evidence, using 5-HT agonists and antagonists, for a specific role for serotonin signaling in guiding regenerating RGC axons after optic nerve injury. The authors establish that this role is specific to regenerating axons and is not essential for developmental RGC axon growth. The authors also demonstrate how their previously established larval zebrafish model for optic nerve injury can be utilized for drug screening to identify pharmaceuticals that impact the normally successful regeneration of CNS nerves in zebrafish. Both the finding for an *in vivo* role of 5-HT in modulating optic nerve regeneration, and the demonstration of the utility of the model for pharmaceutical screening are significant contributions to the field. The data presented support the conclusion of a role for 5-HT signaling in guidance of RGC axons to the

chiasm during optic nerve regeneration and lend support to the notion that axon regeneration is not simply a recapitulation of developmental axon growth. I recommend a few minor revisions that will strengthen the clarity of the report.

Comments for the author

1. Out of the 1400 compounds, was reinnervation only impacted by 5-HT receptor agonists or antagonists? It would be nice to have a table of at least the 40 gene4c pathways.
2. Figure 1 D - I am confused about the orientation of the image of 48 hpt tecta. I think this image inverted, or is it indicating that the reinnervation is happening primarily in the ipsilateral tecta?
3. Figure 4C - is there a way to determine if WAY-100635 exposed nerves are same diameter? From the representative image it looks like growth may be less robust? The conclusion that WAY-100635 does not promote ectopic growth is supported, but there should be some mention that more detailed studies would need to be carried out to establish that it has no effects on development.
4. A more descriptive label for the Y axis in graphs 2D, E, F; 5D, E; Supp 2 G, H would help readers understand what is represented. Instead of "ratio of nerves", perhaps something like "relative rate of reinnervation of contralateral tecta" or "relative rate of ectopic growth". A more detailed explanation of the term should then be included in the figure legend. Figures and legends should stand alone without referring to the Materials.

First revision

Author response to reviewers' comments

Comments from the Reviewers:

Reviewer 1: SUMMARY OF THE ADVANCE MADE IN THIS PAPER AND ITS POTENTIAL SIGNIFICANCE TO THE FIELD

This manuscript reports a previously uncharacterized function of serotonin type-1 receptors (5-HT1) in optic nerve (ON) regeneration in vivo. Using an elegant ON transection assay in zebrafish combined with a small molecule library screen, the authors identified several compounds that modulate serotonin signaling and modified ON regeneration. The authors then demonstrate that 5-HT1 receptors are expressed by retinal ganglion cells (RGCs) before transection and during ON regeneration, and that inhibiting 5-HT1A receptors reduces regeneration but does not affect developmental axon outgrowth. The authors finally show that activating 5-HT1B/D receptors can enhance regeneration. Overall, this study makes a significant contribution to the field by uncovering a novel and specific function of 5-HT1 receptors in ON regeneration.

SUGGESTIONS TO AUTHORS

The study is well carried out, technically of high standard, and provides a thoughtful discussion about the functions of serotonin signaling in the regulation of nerve regeneration. It is appropriate for publication in Development provided the authors address the concerns detailed below:

- 1) One major concern is the use of a single 5-HT1A antagonist (WAY-100635) and a single 5-HT1B/1D agonist (zolmitriptan) to define the role of 5-HT1 receptors in ON regeneration. Different dose-dependent responses were observed for each treatment, which might be explained by different affinities of the drugs for their targets. For instance in figure 2D, 5 μ M of WAY-100635 seems to cause a decrease (although not significant) in ratio of nerves that contrasts with the strong increase observed with 50 μ M. Since WAY-100635 also has a high affinity for D4 receptors, its effects should be confirmed with a second, independent 5-HT1A antagonist or by analyzing *htr1aa/ab* crispants. Similarly, the functions of 5-HT1A, 5-HT1B and 5-HT1D should be further examined using a 5-HT1A agonist and 5HT1B/1D antagonists.

-We enthusiastically thank reviewers for reading, analyzing, and sharing their suggestions and

comments on the manuscript. We appreciate the suggestion of confirming the effects of the 5-HT1A antagonist WAY-100635 in optic nerve regeneration. To address this concern, we successfully created stable CRISPR-Cas9 fish lines to knock out the function of the *htr1aa* and *htr1ab* genes in vivo. Using this approach, we created the following stable mutant line: *Isl2b:GFP; htr1aa +/-; htr1ab +/-* (both mutations predicted to be loss-of-function mutations). Next, we performed the following cross to obtain *htr1aa; htr1ab* double mutants: *Isl2b:GFP; htr1aa +/-; htr1ab +/-* X *Isl2b:GFP; htr1aa +/-; htr1ab +/-* fish. The progeny of this cross was raised to 5 dpf, and an optic nerve transection assay (see details of procedure in material and methods and in the main text), as described in the manuscript, was performed to determine the role of *htr1aa/htr1ab* double mutants in optic nerve regeneration. Unfortunately, after performing multiple rounds of this experiment, we could not obtain enough double mutant progeny due to an increase in lethality during development and the larvae not surviving through the optic nerve procedure. Given the difficulty and complexity of the genetic experiments outlined above, these experiments are out of the current scope of the questions we set out to answer for this manuscript. Similarly, we appreciate the suggestions to examine the effects of 5-HT1A agonists and 5-HT1B/1D antagonists in optic nerve regeneration. While the results obtained after testing these small molecules could lead to a better understanding of the intricacies of these receptors in optic nerve regeneration, we do not expect the findings to significantly affect the conclusions we have posited in our manuscript. Therefore, we believe the small molecule experiments suggested here are out of the scope of this manuscript.

2) Figure 2F: the authors applied WAY-100635 at 32 hpf to assess the effects of 5-HT1A inhibition after axons have reached the optic chiasm. Where are axons at 32 hpt (or just after 32 hpt when the antagonist is active)? Have they crossed the chiasm and already entered the optic tract, or have they just reached the chiasm but not crossed it yet? It would be interesting to clearly pinpoint when, during their navigation, RGC axons lose their sensitivity to the antagonist. Adding a picture of RGC axons at 32 hpt might clarify this point.

-Thank you for this important suggestion. We also believe it is important to show a representative image of the trajectory of regenerating RGC axons at 32 hours post-transection in wild type (DMSO-treated) animals. To address this suggestion, we have obtained images at this time point and have modified Figure 1 to include fluorescent images of the chiasm and tectal region at 32 hpt. **Please refer to the new Figure 1D, the main text and Figure 1 legend for more details and descriptions on where are regenerating RGC axons located at this time point.** We found that at 32 hpt, a small group of regenerating RGC axons have reached the midline (Figure 1D, top panel). However, these axons have yet to reach and re-innervate the tectal region. Degenerated tectal regions are outlined with dashed lines (Figure 1D bottom panel). Note: when obtaining confocal images of the optic tectal region at 32 hpt, we were unable to obtain an image that did not have a significant amount of skin cells overlapping the tectal region. Therefore, we decided to remove from our Z-stack projection any images where the skin overlapped with the tectal region so that readers can easily observe how the tectal region looks at this time point. **For full transparency, we have included the original unaltered z-stack below.** If the reviewers or editors believe additional information should be added to the figure legends to describe the processing outlined above, we are happy to do so.

Unaltered z-stack

Modified z-stack with skin removed, rotated image

3) Figure 3 shows a clear expression of *htr1aa* and *htr1ab* in RGCs in the uninjured retina and at 24 hpt. Are *htr1aa* and *htr1ab* still expressed after 32 hpt, when application of WAY-100635 does not have any effect on regeneration?

-Similar to the previous suggestion, this is a crucial point to address and an experiment that should be included in our manuscript. Therefore, we performed a fluorescent *in-situ* hybridization experiment using the same *htr1aa* and *htr1ab* probes shown in Figure 3 to determine whether these receptors are also expressed at 32 hpt. **The manuscript now includes a Supplementary Figure 3, which includes images of the expression of *htr1aa* and *htr1ab* at 32 hpt.** We found that in contrast to the staining pattern observed at 24 hpt, no specific staining was observed in RGCs after treating samples with *htr1aa* and *htr1ab* probes at 32 hpt. Please refer to Supplementary Figure 3, the main text, and the figure legend for more details and descriptions of this new figure. The main text has been edited to say: “However, we did not observe any staining after incubating transected larvae with probes against these receptors during regeneration at 32 hpt (compare Figures S3A-C and Figures S3D-F).”

4) The authors provide a nice discussion on the possible mechanisms whereby 5-HT₁ receptors regulate ON regeneration. Is serotonin observed along ON tracts after transection? Can 5-HT₁ receptors be detected along regenerating axons (it might be difficult to answer this last question considering the paucity of antibodies working in zebrafish)

-Thank you very much for the suggestion. We agree that it would be valuable to determine whether serotonin is present along the ON tracts after transection. While anti-5-HT antibodies have been used to stain zebrafish neurons in the past, staining the CNS and optic tract after transection has proven to be a difficult task due to the limited and ineffective nature of antibodies in the zebrafish system. Given that the answer to the scientific question posed by our reviewer should not greatly affect the conclusions of our study, we believe this experiment should be addressed in a future publication.

5) The authors indicate in Table 1 that applying serotonin reduced tectal re-innervation, which contrasts with the effects of zolmitriptan and WAY-100635. Are there other serotonin receptors expressed in the visual system that could explain this result?

-The reviewer makes an excellent observation about the phenotype observed when applying serotonin to transected larvae that contrasts with the phenotypes observed when treating larvae with small molecules that target 5-HT₁ receptors. As they mentioned above, we also hypothesize that this contrast in phenotype is due to the expression/function of additional 5-HT receptors during optic nerve regeneration. There is a publication (PMID: 33357413) that looked at the

expression of genes in RGCs at 5 dpf (the time point during which we perform our transection assays) using single-cell transcriptomics. Interestingly, Kolsch *et al.* 2021 found that both *htr1aa* and *htr1ab* genes are expressed in multiple clusters/RGC cell types at 5 dpf. Their results validate our expression data for the *htr1aa* and *htr1ab* genes. No other serotonin receptor was significantly expressed at 5 dpf according to the Kolsch *et al.* database. However, their database shows that the gene *htr2cl1* is differentially expressed in the retina of adult zebrafish.

Two additional publications show expression of different 5-HT1 receptors in larval zebrafish. Norton *et al.*, shows expression of *htr1d* in the retina of 3 dpf zebrafish larvae while Pei *et al.*, shows expression of *htr1b* in the retina of 2 dpf larvae. In summary, while there is not a specific study that shows the expression of additional serotonin receptors at the time point when our optic nerve transections are performed, given the expression data mentioned above and our functional studies, it is plausible that the *htr1b*, *htr1d* or different serotonin type-1 receptor(s) could be expressed at this time point, leading to the phenotype observed when treating with serotonin. Finally, we applied serotonin at a 10 μ M concentration (same concentration used for the rest of the small molecules in our drug screen). Therefore, applying serotonin at different concentrations could lead to different phenotypes, including ones that more closely resemble the phenotypes observed when treating with the *htr1b/1d* agonist.

Minor points:

> In material and methods, please clarify the formula used to calculate ratios for drug treated groups: what happens to the ratio if the # of [control treated] ON nerves with defect is zero (does this happen)? Also, isn't "ON nerves" redundant?

-We thank the reviewer for their valuable feedback on our Material and Methods. To clarify the formula, the reviewer brings up an excellent point about the possibility of an experiment in which the # of [control treated] ON nerves with defects is zero. Before performing the experiments, we considered this possibility and decided we would standardize our variables to create variables that are plurality measures (Wexler *et al.*, 2017) if any result ended with zero as the denominator. After performing the ON nerve experiments, we observed that none of our results showed the # of [control treated] ON nerves with defects as zero. Therefore, we elected not to standardize the data and instead insert the raw data in our formula in favor of transparency. Lastly, we agree that 'ON nerves' is redundant wording in the formula, and so we decided to remove the phrase from the formula.

> Figure 3: please indicate whether optical sections are coronal or longitudinal.

-Thank you for your comment on this figure. We have updated the figure legend of Figure 3 (and all other figure legends showing images with Z-projections). The figure legend(s) now read "maximum Z-projection of X **horizontal** optical sections..."

> In the text describing Figure 4, the conclusion should be that 5-HT1A (not 5-HT1) receptor function is dispensable for RGC axon outgrowth. Since WAY-100635 is specifically targeting 5-HT1A receptors, a role for 5-HT1B or 5-HT1D cannot be ruled out.

-The reviewer is correct in respects to the conclusion drawn for Figure 4. We have updated the manuscript to say "...we conclude that **5-HT1A** receptor function is likely dispensable in development for RGC axons to project to the optic tectum".

> Panels A-F' in Figure S2 should be combined with Figure 3 (I wondered what the expression of *htr1b* and *htr1d* was before finding out the answer later in the paper). Panels G and H could then be added to Figure 5.

-We appreciate the reviewer's suggestion of combining Figure 3 and some panels part of Supplementary Figure 2. We would prefer to keep the current figure organization and structure because by adding eight additional panels to Figure 3, we risk diluting the main findings and conclusions of our expression data. Moreover, while these supplemental panels are not part of Figure 3, their descriptions and citations can be found in the same paragraph where the findings

related to Figure 3 are described.

The explanation above also applies to our organization and structure of Figure 5.

> Figure 5: Showing pictures illustrating the effects of both concentrations of zolmitriptan would strengthen the study. Also, the manuscript indicates that panels A-D correspond to 5 μ M of zolmitriptan, but the legend indicates 50 μ M.

-We thank the reviewer for suggesting including an image of the phenotype observed after treating transected larvae with 5 μ M of Zolmitriptan. We performed an optic nerve transection experiment and treated larvae with 5 μ M of Zolmitriptan at 24 hpt. Please see below for three representative images showing optic nerve regeneration at 48 hpt:

Figure for Reviewer: 5 μ M Zolmitriphan treatment does not affect RGC regeneration (3 samples)

Image Legend:

Fluorescent representative image of three different *Tg(Isl2b:GFP)* larvae (A, A' and A'') at 48 hpt treated with 5 μ M Zolmitriptan. Regenerating optic nerve of larva treated with 5 μ M Zolmitriptan shows RGC axons regrowing toward the optic chiasm (top panel) and how they begin to re-innervate the peripheral edges of the optic tecta (bottom panel). The phenotype observed when treated with this agonist concentration is indistinguishable from wild-type animals. Therefore, we believe these images do not need to be included in our figures. Reviewer 2: The authors investigate the role of serotonin (5-HT) signaling in optic nerve regeneration using larval zebrafish as a model. They conducted a small molecule screen and identified serotonin type-1 (5-HT1) receptor modulators that impact optic nerve regeneration. Their findings suggest that inhibiting 5-HT1 receptors or components of the 5-HT pathway selectively impedes regeneration, while agonist-mediated activation enhances axonal regrowth. The authors conclude that 5-HT1 receptors play a critical role in promoting early stages of optic nerve regeneration and propose that serotonin-dependent neuromodulation directs optic nerve regeneration in vivo. The manuscript provides preliminary insights into the role of serotonin signaling in optic nerve regeneration in zebrafish. However, the study has several critical flaws and would benefit from more evidence to support the claims. To be suitable for publication, the

paper requires extensive revisions and substantial additional work.

*We want to thank the reviewer for taking the time thoroughly read, review and analyze our manuscript and share their comments and detailed suggestions. Please see below for our responses, changes and outlook on each of these points.

I am particularly concerned about the following points:

- The manuscript relies heavily on pharmacological inhibition and activation of serotonin receptors using small molecules. While this approach is valuable for initial screening, it is insufficient to establish a direct, specific role for serotonin receptors in optic nerve regeneration. The use of complementary genetic approaches, such as CRISPR/Cas9-mediated knockouts, RNA interference, or transgenic overexpression of specific serotonin receptor subtypes (e.g., 5-HT1A, 5-HT1B) in zebrafish, would provide more direct evidence for the role of these receptors. For example, the manuscript claims that 5-HT1 receptor signaling is critical for axonal regrowth but fails to distinguish whether this effect is due to receptor activity in RGCs or other cell types. Genetic tools could allow for cell type-specific manipulation, thereby strengthening the claims of receptor involvement in RGC axonal regeneration.

*We thank the reviewer for all the insightful suggestions outlined here to improve our manuscript and investigation. We agree that the genetic approaches highlighted above, including CRISPR/Cas9-mediated knockouts and transgenic overexpression of different serotonin receptors, will undoubtedly be important towards a full understanding of the role of serotonin in CNS regeneration. To provide additional evidence for the role of 5-HT1 receptors, we created stable CRISPR-Cas9 fish lines to knock out the function of the *htr1aa* and *htr1ab* genes *in vivo*. We successfully created stable mutant lines using this approach, allowing us to test the following loss-of-function double mutant: *Isl2b:GFP+*; *htr1aa* *-/-*; *htr1ab* *-/-* and compare their phenotype(s) with their siblings. The progeny was raised to 5 dpf, and an optic nerve transection assay (see details of procedure in material and methods and the main text), as described in the manuscript, was performed to determine the role of *htr1aa/htr1ab* double mutants in optic nerve regeneration. Unfortunately, after performing multiple rounds of this experiment, we could not obtain enough double mutant progeny due to increased lethality during development and the larvae not surviving through the optic nerve procedure. Given the difficulty and complexity of the genetic experiments outlined above, these experiments are out of the scope of the questions we set out to answer for this manuscript.

We believe that the additional approaches highlighted here, such as the transgenic overexpression of specific serotonin receptor subtypes (including 5-HT1A and 5-HT1B) to distinguish whether receptor activity is in the RGCs or a different cell type, are out of the current scope of this investigation. As mentioned by the reviewers, our findings, while primarily using a pharmacological approach, provide the first evidence for mechanisms through which serotonin-dependent neuromodulation directs optic nerve regeneration *in vivo*. In the future, it will be important to determine whether these 5-HT receptors act autonomously or non-cell autonomously with respect to RGCs.

- The manuscript describes a small molecule screen using a library of FDA-approved compounds to identify modulators of serotonin signaling. However, the criteria for selecting the serotonin-related targets are not sufficiently detailed. The authors could clarify how compounds were chosen for follow-up studies, including their specificity, known off-target effects, and pharmacokinetic properties. For example, why were only the four serotonin-targeting compounds selected for detailed study, and what evidence supports their specificity for serotonin receptors in the context of zebrafish? Additionally, details on the validation of hits from the small molecule screen are sparse. The authors could provide a more comprehensive description of the follow-up experiments conducted to confirm the initial screen findings, such as dose-response tests, time-course experiments, and verification in independent replicates.

*We thank the reviewer for their interest in learning more about the FDA-approved library that we used for our drug screen. The main criteria we used for selecting serotonin-related targets, which is the same criteria we used for other pathways analyzed, was to select small molecules that have thorough documentation on the proteins they target *in vitro* and/or *in vivo*. We included information on how we chose small molecules for follow-up studies in the Material and Methods section. To clarify the methods provided, we have edited this section to say (portions in bold are

new edits to be included in the manuscript): “Three unique small molecules were added to ‘small molecule pools’ to determine their effect on optic nerve regeneration. **Usually, each drug pool was tested on six larvae as part of an experimental group. If the result was not clear, the test, including the re-testing of DMSO treated control groups, was repeated. Pools which caused impaired regeneration in more than half of the tested larvae (usually 4 out of 6) were called “hits”.**”

“Investigators were blinded to the identity of the small molecules in each drug pool. Small molecules that lead to lethality, alteration of body morphology, or a significant reduction in responsiveness to touch during the treatment period were excluded from further testing. All small molecules of drug pools that impaired regeneration were then tested individually to validate the phenotypes observed using the assay described above. **Usually, each individual small molecule was tested on eight larvae as part of an experimental group. Pools that caused impaired regeneration in more than half of the tested larvae (usually 6 out of 8) were validated and analyzed further, including the use of dose-response and time-course experiments, many of which were highlighted in this publication.**

Individual small molecules that impaired regeneration were re-tested using a different batch of the same small molecule and confirming they impaired optic nerve regeneration.”

- The authors may consider addressing potential sources of bias or error in the experimental design. For instance, while the small molecule treatments were blinded, it is unclear if the subsequent imaging and quantification were also conducted blind to treatment conditions. Unblinded assessment of regeneration could introduce bias in quantifying regrowth or misguidance. The authors could provide more information on whether all stages of the experiment, including the assessment of axon regrowth and statistical analysis, were blinded and how randomization was achieved. Additionally, there is no discussion of the reproducibility of the findings across different experimental batches, which is critical to ensure the reliability of the reported effects.

*We thank the reviewer for carefully reading and examining our experimental methodology. We are happy to share additional information on how the experiments in this manuscript were performed with the purpose of boosting the reproducibility of the findings. We have edited the main text in the following way to include the requested information (**edits/new information are in bold**): “Three unique small molecules were added to ‘small molecule pools’ to determine their effect on optic nerve regeneration. Investigators were blinded to the identity of the small molecules in each drug pool. **Experimental and control groups were treated in the same order as optic nerve regeneration was assessed to ensure close to equal treatment duration. Assessment of optic nerve regeneration was performed for one group after the other.** Small molecules that lead to lethality, alteration of body morphology, or a significant reduction in responsiveness to touch during the treatment period were excluded from further testing. All small molecules of drug pools that impaired regeneration were then tested individually to validate the phenotypes observed using the assay described above. **The identity of pools and compounds was revealed only after scoring regeneration, reducing subjective the bias.** Individual small molecules that impaired regeneration were re-tested using a different batch of the same small molecule, confirming they impaired optic nerve regeneration. **All zebrafish larvae were randomly distributed into a control and experimental group for each experimental batch.**”

- While the manuscript presents data on the effects of serotonin signaling modulation on optic nerve regeneration, it does not provide functional validation of the proposed mechanisms. For example, the study suggests that 5-HT1 receptors direct regenerating axons toward the optic chiasm by modulating growth cone guidance, possibly via cAMP signaling pathways. However, no direct evidence, such as measurements of intracellular cAMP levels, or experiments manipulating downstream signaling components, is provided to support this mechanism. The authors could consider performing functional assays to validate these claims, such as live-cell imaging of growth cones or molecular assays to quantify changes in second messengers (e.g., cAMP) upon receptor activation or inhibition. This would provide more concrete evidence to support the proposed mechanistic pathways.

*We thank the reviewer for highlighting our discussion on possible mechanistic pathways downstream of the serotonin receptors. The proposed discussion on cAMP levels and other

downstream signaling components is included to reflect on the critical implications our findings could have on known pathways involved in CNS regeneration. We are aware that this is only one possibility of how the serotonin pathway may act mechanistically *in vivo*. To make it clear to our future readers that this is only one possible mechanism we are considering, we have modified the text in the Discussion to say (see edits in bold): “Given the expression of diUerent 5-HT1 receptor genes in RGC neurons during regeneration (this study), a **potential** mechanism is that serotonin **could** act as a guidance cue along the optic nerve tracts and that its binding to 5-HT1 receptors on RGC axons repulses axons away from the retina and towards the optic chiasm. These receptors **may**, in turn, modulate cAMP signaling, which is **known to be** pivotal for promoting RGC axon regeneration **in different models.**”

- The manuscript would benefit from a more comprehensive description of the control experiments used to validate the specificity of the findings. For example, it is critical to demonstrate that the observed effects on optic nerve regeneration are specifically due to the modulation of serotonin signaling and not secondary to general toxicity or oU-target effects. The authors mention using a DMSO control but could also include additional controls, such as vehicle-only controls for each small molecule, and parallel experiments using structurally unrelated inhibitors of serotonin receptors to demonstrate that the effects are specifically mediated through serotonin signaling. Moreover, for the genetic manipulations suggested above, appropriate controls, such as scrambled or non-targeting controls, would be necessary to validate the specificity of the observed phenotypes.

*We thank the reviewer for their comments and suggestions on control groups. We agree that we should have been more descriptive on noting that in each experimental replicate we performed, we always included a minimum of five samples (i.e., 5 optic nerves analyzed during optic nerve regeneration) as part of the DMSO-control groups. This practice allowed us to determine whether the transection and regeneration assay behaved as expected during each replicate. We have added a sentence in our methods to explicitly include this practice as part of our procedure (edits shown in bold): “Compounds were applied to wells in a 48-well plate dish containing fully transected 6 dpf larvae at diUerent time points, as indicated in the results section. Control larvae received a solution with 0.3% DMSO in PTU/E3. **For each individual replicate performed, Isl2b:GFP+ larvae were pooled and randomly assigned to control and experimental groups. Each control pool contained at least five samples (i.e., 5 optic nerves) that were analyzed during optic nerve regeneration.**”

In addition to DMSO controls, while performing the small molecule screen, we also included larvae that were transected and incubated in E3 medium throughout the duration of the experiment. The optic nerve regeneration results obtained from these E3 groups were indistinguishable from the results observed using DMSO control groups.

- The manuscript describes the use of serotonin receptor agonists and antagonists at diUerent concentrations and time points but requires a discussion on the temporal dynamics of serotonin signaling during optic nerve regeneration. The authors claim that serotonin signaling is particularly important during the early stages of regeneration, but the manuscript could include time-course studies to map out when serotonin receptor activity is most critical. To strengthen the findings, the authors could perform and discuss time-course experiments to identify the precise temporal windows when serotonin receptor modulation aUects regeneration. This would provide a more detailed understanding of how serotonin signaling integrates into the broader context of regenerative processes.

*We thank the reviewer for their suggestions on how serotonin signaling is involved at distinct temporal windows throughout the regenerative process. We agree that it is critical to learn more about the role of serotonin modulation in the later stages of optic nerve regeneration. Therefore, we performed the following experiments during optic nerve regeneration at 32 hpt: (1) Analyzed and obtained representative images of the trajectory of regenerating RGC axons at 32 hpt in wild type (DMSO-treated) animals. We obtained images of the optic chiasm and optic tecta at this time point. To include these new images in our manuscript, we have modified Figure 1 to include max-projection images of the chiasm and tecta region at 32 hpt (new Figure 1D).

(2) We performed an *in-situ* hybridization experiment where we stained transected larvae with *htr1aa* and *htr1ab* probes to determine whether these receptors are expressed at 32 hpt. The

findings of this experiment can be found in the new Supplemental Figure 3A - F. We found that in contrast to the staining pattern observed at 24 hpt, no specific staining was observed in RGCs after treating samples with htr1aa and htr1ab probes at 32 hpt. Please refer to Supplementary Figure 3, the main text, and the figure legend for more details and descriptions of this new figure. Based on the results obtained, we have updated our manuscript's main text to say: "However, we did not observe any staining after incubating transected larvae with probes against these receptors during regeneration at 32 hpt (compare Figures S3A-C and Figures S3D-F)."

- The manuscript's methods section could be improved with more and sufficient detail in several areas that are critical for reproducibility and interpretation of the findings. For example, there is insufficient information about how the small molecule concentrations were chosen, how often treatments were refreshed, or how long the treatments were applied relative to key developmental or regenerative milestones. The manuscript could provide a more detailed methodological description, including justifications for all chosen experimental conditions, to ensure that other researchers can replicate the findings.

*We thank the reviewer for their suggestions on how to improve our methodology section. We are happy to clarify or expand on the sections the reviewer highlighted here to improve the reproducibility and interpretation of our findings. For the small molecule screen, the standard concentration of 10 μ M was chosen as a baseline for all the small molecules that were initially tested because a previous report using the same Drug Library: 'Selleckchem Bioactive: 2100 FDA-approved/FDA-like small molecules' used the 10 μ M concentration as a baseline and found a small amount of lethality (less than 5%) and a significant amount of hits in relation to their neurobiological phenotype of interest. To reflect this reasoning, we have edited our Material and Methods section (**edits in bold**): "Stock and working small molecule solutions were prepared as described previously (Lamire et al., 2023)".

Briefly, stock solutions (100x frozen stocks in DMSO) were initially diluted 1:100 in E3, obtaining a 10x solution. 30 μ L of this solution was then added into the wells, yielding a 10 μ M drug concentration in 0.3% DMSO.

Experimental procedures varied on whether the small molecule administered needed to be refreshed or not. The experiment that required small molecules to be refreshed is described in Figure 2G. This figure includes an experimental procedure timeline that details at which time points the drug was replenished. Therefore, we believe no additional edits are required to our text.

As for the duration of the small molecule treatments in each experimental procedure, we agree that this information may not be as clear in our Methods and Figure Legends. To clarify how long the treatment(s) were applied, we have modified the Methods section (**edits in bold**): "Larvae that displayed no axonal-GFP remnants from the injury site to the tectum were added to 48-well plates and treated with DMSO 0.3% or small molecules **from the 24 hpt to 48 hpt time points**, unless otherwise noted. Additionally, wherever relevant, figure legends from Figures 2, 4, and 5 will be modified to clarify the duration of the small molecule treatment. For example, the figure legend of Figure 2D now reads: "Quantification of optic nerve axonal re-innervation to contralateral tectum at 48 hpt in *Tg(Isl2b:GFP)* larvae treated **from 24 hpt to 48 hpt** with DMSO 0.3% or WAY-100635."

- The authors use small molecule inhibitors and agonists, which may have off-target effects that could confound the interpretation of the results. For instance, the use of WAY-100635, a known 5-HT_{1A} receptor antagonist, may also have affinity for other receptors or signaling pathways. The manuscript could discuss these potential off-target effects and consider performing additional experiments, such as using more selective inhibitors or genetic approaches, to confirm that the observed effects are truly due to serotonin receptor modulation.

*We agree with the reviewer that discussing drug specificity would be useful. Therefore, when introducing the small molecule WAY-100635, we have added the following sentence (edits are highlighted in bold): "...5-HT_{1A} receptors in vitro, with 100-fold higher binding selectivity to 1A receptors over other subtypes **and shows lower affinity to dopamine D₄ receptors (Chemel et al., 2006)**".

When introducing Zolmitriptan, we have added the following sentence (edits are highlighted in bold): "To test this, we used the 5-HT_{1B/1D} selective agonist Zolmitriptan, **with modest**

affinity to 5-HT1A receptors...". Please see above for a discussion on our different approaches to validate our pharmacological manipulations.

Major comments:

- The manuscript would be stronger if it considered the relevant literature, especially studies related to serotonin receptor signaling in CNS regeneration, including brain regeneration in zebrafish and other models. For example, the claim that this study is the first to define an acute, in vivo role for serotonin receptors in CNS axon regeneration is not accurate. Several studies have already shown relevance of serotonergic signaling for CNS axon regeneration (PMID 34876587, 31905199, 26565906 and others). The authors must provide a more thorough and balanced discussion that includes studies supporting and contradicting their findings.

*We thank the reviewer for their comments on the role of serotonin in CNS regeneration and for sharing relevant literature and citations in this topic. The author is correct when they state that our study is not the first to define 'an acute, in vivo role for serotonin receptors in CNS axon regeneration'. However, the claim that we are making in our manuscript is that 'the findings reported here are the first to define an acute, in vivo role for serotonin receptor signaling in **long-distance CNS axon** and optic nerve regeneration. The phrase 'long-distance' is crucial to our claim and we believe that, to the best of our knowledge, this statement is true.

To be fully transparent with the reviewer, we read and analyzed the citations provided here and decided whether these citations should be included in our manuscript. We mentioned and cited PMID 34876587 in our Introduction: "Finally, serotonin receptor signaling has also been implicated in post-developmental processes, including axonal regrowth.

Following a spinal cord transection in adult zebrafish, 5-HT1B receptors facilitate the regeneration of spinal interneurons (Huang et al., 2021)." No changes to the text were made.

We agree with the reviewer that the main findings in PMID 26565906 should be discussed and cited in the Discussion section of our text. We have modified our main text to the following (**edits in bold**): "**Previous studies in adult teleost fish have shown that after spinal cord injury, serotonin promotes motor neuron regeneration (Barreiro-Iglesias et al., 2015), as well as facilitate axonal regeneration of local spinal interneurons.**"

Finally, while PMID 31905199 shows how the Serotonin-BDNF-NGFR axis enables regenerative neurogenesis in Alzheimer's, their work did not focus on the role of serotonin in long-distance CNS axon regeneration. Therefore, we decided not to include this citation in our manuscript.

- The localization of serotonin receptors is not clearly demonstrated in the manuscript. The in situ hybridization is vague and combines probes, making it difficult to confirm the specific expression of these receptors in retinal ganglion cells (RGCs). The authors could utilize existing single-cell RNA sequencing (scSeq) datasets for zebrafish to externally validate their claims about receptor expression, as this is crucial to substantiate the specificity of their findings.

*We would like to thank the reviewer for sharing their knowledge on these existing scSeq datasets for zebrafish. As the reviewer points out, there is a publication (PMID: 33357413) that looked at the expression of genes in RGCs at 5 dpf (the time point during which we perform our transection assays) using single-cell transcriptomics. Kolsch et al. 2021 found that both *htr1aa* and *htr1ab* genes are expressed in different clusters/RGC cell types at 5 dpf. Therefore, their results validate our in-situ hybridization expression data for the *htr1aa* and *htr1ab* genes. Importantly, no other serotonin receptor was significantly expressed at 5 dpf according to the Kolsch et al. database. We have now referenced and cited Kolsch et al., in our manuscript. We have also modified the main text of our manuscript in the section related to Figure 3 to state (**edits in bold**): "5-HT1 receptors are expressed in the vertebrate CNS, including in the retina of mice and zebrafish larvae (Upton et al., 1999; Norton et al., 2008). Moreover, an existing single-cell transcriptomics (scrNA-seq) dataset used to generate the molecular taxonomy of RGCs in zebrafish showed that 5-HT1A receptors are expressed in a subset of RGCs in the 5 dpf zebrafish larvae (Kolsch et al., 2021)."

- It would be useful to consider the pleiotropic effects of serotonin inhibition. Given that the entire larvae are treated with antagonists or agonists, the specificity of the observed effects on optic nerve regeneration is questionable. The authors must address the potential for broad, non-specific effects on multiple serotonin pathways and provide a detailed discussion of these limitations.

*Please see below for our comments on the limitations of the study, in relation to the specificity of our small molecules and their effects on optic nerve regeneration.

- The figures need essential details, such as individual data points and distribution, which prevents a clear understanding of data variance. The authors could revise the graphs to include individual data points and provide a more detailed description of the variance in their data. Additionally, the manuscript will benefit from a discussion on the appropriateness of the statistical methods used. For example, it is unclear if the sample sizes are sufficient for the conclusions drawn, and whether proper statistical controls were employed.

*We thank the reviewer for evaluating our choice for data presentation with the goal of making our figures and analysis stronger for publication. While we appreciate the suggestion, we believe that in the case of the optic nerve regeneration data presentation, using the formula included in the Material and Methods section ‘Quantification of Optic Nerve (ON) Regrowth Phenotypes’, which normalizes the data obtained from the experimental groups to the control groups, provides relevant information on data variance. Moreover, we are happy to include additional data used to generate the graphs in our manuscript if the editor believes it is appropriate.

- The quantification approach for spinal cord regeneration in the manuscript is not well justified. The authors observe no change in regeneration but may be using an inappropriate readout. The authors could consider alternative readouts or regions to quantify regenerative changes, possibly by comparing with established models or metrics used in similar studies. Several studies showed changes in axon regeneration in spinal cord with serotonin modulation.

*We thank the reviewer for their comments on our results related to spinal cord regeneration. While it is true that studies have shown changes in spinal cord regeneration with serotonin modulation, we are not aware of a previous study that shows serotonin modulation affecting long-range CNS regeneration in zebrafish larvae. Therefore, we used the Mauthner regeneration paradigm (Figure 2G) to test the role of 5-HT₁ receptors in long-range CNS regeneration. The methodology and statistical analysis used in this experiment has been reported in multiple publications (please see Bremer et al., 2019, Meserve et al., 2024), arguing that this is a proven experimental method that yields significant results when analyzing a pathway that is involved in this regeneration process.

- The observed differences between high and low doses of antagonists suggest pleiotropic effects of the treatments. It would be beneficial if the manuscript provided a dose-response curve and discussed the implications of the pleiotropic effects of their treatments.

*We thank the reviewer for this comment and refer them to the last section of our discussion, where we discussed in detail the different concentrations of the antagonist and agonist and how they lead to different effects on the observed phenotype(s):

“On the other hand, an increase in 5-HT₁ receptor signaling, similar to the levels attained after ectopic activation with 5 μ M of the agonist zolmitriptan, leads to enhanced optic nerve regrowth, culminating in higher rates of innervation to the contralateral tectum.

Finally, increasing 5-HT₁ receptor signaling levels further with 50 μ M of zolmitriptan causes significant ectopic optic nerve regrowth, possibly due to a 5-HT₁ receptor-mediated increase in the repulsion of axons to serotonin.”

While we report on the findings for varying concentrations of WAY-100635 in the results section: “Compared to DMSO-treated control animals, we observed a significant and dose-dependent reduction of optic nerve regrowth in WAY-100635 treated larvae (compare Figures 2A-D, DMSO controls with 5 μ M or 50 μ M) WAY-100635 treated larvae”, we decided not to discuss the effects of the lower concentration of WAY-100635 in optic nerve regeneration in the Discussion section because it did not reach statistical significance when compared to controls.

- The manuscript should contain a paragraph discussing the limitations of the study, which is crucial in this case. The authors could add a comprehensive limitations section to acknowledge these issues and provide context for the interpretation of their findings.

*We thank the reviewer for pointing out here and in other comments their thoughts and suggestions on the limitations of our study. As it is currently written, we have not included a

specific section that focuses on the limitations of the study. Rather, we have highlighted different limitations of our study throughout our manuscript where we believe it is pertinent based on the experimental analyses performed and discussed. See below for an example of a limitation we highlighted in our Discussion (**in bold**): “It is therefore tempting to speculate that this differential requirement simply reflects a higher level of genetic redundancy among 5-HT receptors during development compared to regeneration. [...] **Another possibility is that a different set of 5-HT receptors might function during optic nerve development. In fact, there are over 20 serotonin receptors in the zebrafish genome, most of which have their expression and function yet to be explored.**

Therefore, a thorough combinatorial analysis of 5-HT receptors will be critical to determine whether 5-HT receptors play a redundant role in optic nerve development and regeneration.” We are happy to include additional limitations and/or a limitations section if the editor deems it necessary.

Given the preliminary nature of the findings and the substantial revisions needed, I recommend a major revision. The authors must address the issues raised above and provide additional experimental data to support their claims more convincingly.

- Authors could expand the literature review. They could include recent studies on serotonin receptor signaling in CNS regeneration, particularly those that both support and contradict the findings of this study. Acknowledging the existing work and discussing how this study adds to or differs from previous research would make sense.

*Please see above for our comments and the citations added to the revised manuscript.

- Authors could clarify the receptor localization. They can perform additional experiments or use available single-cell RNA sequencing datasets to validate the expression of serotonin receptors in the specific neuronal populations of interest. This will strengthen the evidence for receptor involvement in optic nerve regeneration.

*Please see above for our comments and the text edits added to the revised manuscript.

- Discussing the drug specificity and pleiotropic effects would be useful. Authors can provide a more in-depth discussion of the potential pleiotropic effects of the serotonin receptor modulators used in this study.

*Please see above for our manuscript edits and discussion on drug specificity.

- Including a section for limitations of this study will be highly beneficial.

*Please see above for our comments on the limitations of the study.

- Authors could revise data presentation by updating the figures to show individual data points and clearly present the variance. If needed, they could reconsider the statistical considerations, such as reviewing and potentially revising the methods used for quantifying regeneration, especially in the spinal cord regeneration assays. Providing scientific reasoning for the choice of readout and consider alternative methods or areas for quantification will help the audience by reducing the guesswork.

*Please see above for our discussions on data presentation, variance and statistical analyses for both optic nerve and mauthner regeneration.

- Performing a dose-response analysis will strengthen the conclusions. This can also help a discussion on how varying concentrations may lead to different effects and what this implies for the specificity of the observed phenomena.

-Please see our comments above, detailing how we addressed the dose-response analysis suggestions by the reviewer.

Reviewer 3: SUMMARY OF THE ADVANCE MADE IN THIS PAPER AND ITS POTENTIAL SIGNIFICANCE TO THE FIELD

The manuscript "Serotonin neuromodulation directs optic nerve regeneration" provides evidence, using 5-HT agonists and antagonists, for a specific role for serotonin signaling in guiding

regenerating RGC axons after optic nerve injury. The authors establish that this role is specific to regenerating axons and is not essential for developmental RGC axon growth. The authors also demonstrate how their previously established larval zebrafish model for optic nerve injury can be utilized for drug screening to identify pharmaceuticals that impact the normally successful regeneration of CNS nerves in zebrafish. Both the finding for an *in vivo* role of 5-HT in modulating optic nerve regeneration, and the demonstration of the utility of the model for pharmaceutical screening are significant contributions to the field. The data presented support the conclusion of a role for 5-HT signaling in guidance of RGC axons to the chiasm during optic nerve regeneration and lend support to the notion that axon regeneration is not simply a recapitulation of developmental axon growth. I recommend a few minor revisions that will strengthen the clarity of the report.

SUGGESTIONS TO AUTHORS

1. Out of the 1400 compounds, was reinnervation only impacted by 5-HT receptor agonists or antagonists? It would be nice to have a table of at least the 40 genetic pathways.

-We thank the reviewer and appreciate their interest in learning more about the small molecule screen our group performed. To provide more information about the types and example of pathways included in these 40 genetic signaling pathways, we have modified the main text to state the following (edits in **bold**): “..including agonists and antagonists targeting **growth factor pathways, neuronal cell-surface receptors and neurotransmitter systems.**”

2. Figure 1D - I am confused about the orientation of the image of 48 hpt tecta. I think this image inverted, or is it indicating that the reinnervation is happening primarily in the ipsilateral tecta?

-We thank the reviewer for this question. Yes, the ipsilateral tecta are re-innervated to a higher extent at 48 hpt. However, this difference is not observed in wild-type animals by 72 hpt (see Harvey *et al.*, 2019, for more information on that time point). To be transparent with the readers, we have included the following sentence in the Figure 1E legend: “**At 48 hpt, the ipsilateral tectum (left tectum) is typically more innervated than the contralateral tectum.**”

3. Figure 4C - is there a way to determine if WAY-100635 exposed nerves are same diameter? From the representative image it looks like growth may be less robust? The conclusion that WAY-100635 does not promote ectopic growth is supported, but there should be some mention that more detailed studies would need to be carried out to establish that it has no effects on development.

-We thank the reviewer for their thorough analysis of the data and images provided for this figure. After observing the morphology and innervation patterns of all samples in the different experimental groups, we did not see any significant pattern that would lead us to conclude that axonal growth robustness was affected during development. However, we agree that more detailed studies should be performed in the future to establish that 5-HT₁ receptors have no effects in development. We have edited the main text of the manuscript in the results section to state the following (edits in **bold**): “5-HT₁ receptor signaling is **likely** dispensable for developmental RGC axonal growth”. We have added the adverb likely to be transparent about the conclusion reached based on our current analysis.

Moreover, we have a paragraph in the discussion that discusses the possibility of additional 5-HT receptors playing a role during development: “Another possibility is that a different set of 5-HT receptors might function during optic nerve development”.

4. A more descriptive label for the Y axis in graphs 2D, E, F; 5D, E; Supp 2 G, H would help readers understand what is represented. Instead of “ratio of nerves”, perhaps something like “relative rate of reinnervation of contralateral tecta” or “relative rate of ectopic growth”.

A more detailed explanation of the term should then be included in the figure legend.

Figures and legends should stand alone without referring to the Materials.

-We thank the reviewer for their constructive feedback. We completely agree that it would improve our figures and graphs to have our Y axis be more descriptive of the analysis that was performed in each experiment. If the reviewer does not object from us using the exact phrasing they suggested here, we will change our Y axis to state: “**relative rate of ON nerves**”

reinnervation to contralateral tecta” or “relative rate of ON nerves ectopic regrowth” in the appropriate figures. We will also include a more detailed description of each term in the Figure legends as needed. An example of this detailed explanation is described here (**edits in bold**): “**Bars represent the relative rate of optic nerves’ ectopic regrowth** in WAY-100635 treated (black bars) and control groups, plotted as a log₂ scale.”

Second decision letter

MS ID#: dev.204334R1

MS TITLE: Serotonin neuromodulation directs optic nerve regeneration

AUTHORS: Kristian Saied-Santiago; Melissa Baxter; Jaffna Mathiapparanam; Michael Granato

Dear Kristian and Michael,

Apologies for the delay in obtaining reviews but I have now received all the referees reports on the above manuscript, and have reached a decision. The referees' comments are appended below, or you can access them online: please go to .

Although several concerns are raised in the reviews (particularly from referee 2), I think most should be straightforward to address and would like to publish a revised manuscript in Development, once you have done this. Reviewer 2 is the most critical and I don't expect you to perform further experiments to address their concerns but please do modify the text should this help address the points raised.

Please attend to all of the reviewers' comments in your revised manuscript and detail them in your point-by-point response. If you do not agree with any of their criticisms or suggestions explain clearly why this is so. If it would be helpful, you are welcome to contact us to discuss your revision in greater detail. Please send us a point-by-point response indicating your plans for addressing the referees' comments, and we will look over this and provide further guidance.

Reviewer 1

Advance summary and potential significance to field

This manuscript reports a previously uncharacterized function of serotonin type-1 receptors (5-HT₁s) in optic nerve (ON) regeneration in vivo. Using an elegant ON transection assay in zebrafish combined with a small molecule library screen, the authors identified several compounds that modulate serotonin signaling and modified ON regeneration. The authors then demonstrate that 5-HT₁ receptors are expressed by retinal ganglion cells (RGCs) before transection and during ON regeneration, and that inhibiting 5-HT_{1A} receptors reduces regeneration but does not affect developmental axon outgrowth. The authors finally show that activating 5-HT_{1B/D} receptors can enhance regeneration. Overall, this study makes a significant contribution to the field by uncovering a novel and specific function of 5-HT₁ receptors in ON regeneration.

Comments for the author

The study is well carried out, technically of high standard, and provides a thoughtful discussion about the functions of serotonin signaling in the regulation of nerve regeneration. The additional experiments showing that some regenerating axons have reached the midline at 32 hpt and that *htr1aa/ab* receptors are no longer expressed at that timepoint reinforce the conclusions of the study.

Reviewer 2*Advance summary and potential significance to field*

This study provides valuable insights into optic nerve regeneration by highlighting the role of serotonin signaling. The use of zebrafish as a model, coupled with pharmacological screening, contributes to understanding CNS regenerative mechanisms. The findings have potential significance in broader neuroregeneration research, particularly in CNS repair strategies. The authors have addressed several concerns.

Comments for the author

That said, a few critical issues remain unresolved, and I would encourage the authors to engage with them directly to further enhance the clarity, rigor, and impact of their study.

Essential revisions:

1. statistics:

The omission of individual data points remains a significant concern, as it obscures variance and prevents a complete assessment of data distribution. This is not a subjective preference but a requirement stated in the journal's guidelines (Journal's checklist

<https://journals.biologists.com/DocumentLibrary/DEV/Checklist.pdf> states clearly:

Graphs should allow the reader to see the true data spread (i.e., box-and-whisker plots, SuperPlots, etc.). For small sample sizes, individual data points should be plotted.

Without individual data points, key statistical properties such as variance, skewness, kurtosis, and F-values cannot be fully assessed. Statistical summaries, while useful, are not a substitute for raw data visualization. Moreover, no raw data file is provided, making independent verification impossible. Since presenting individual data points is an explicit requirement of the journal, the authors should include them. If there is a compelling methodological reason for not doing so, this should be transparently stated rather than framed as a mere "choice of data presentation."

2. long-range

The distinction between "long-range" and "short-range" regeneration remains unclear. The term does not appear in the title or abstract, was not there in the initial submission, suggesting it was introduced post hoc.

If the authors classify optic nerve regeneration as "long-range," then what constitutes "short-range" in this context? Given that peripheral nerves and some spinal cord models regenerate over even longer distances, it is unclear why this term is relevant. Can the authors provide a biologically meaningful definition rather than using this term as a general rhetoric justification?

3. limitations

The study lacks a dedicated limitations section. While alternative explanations are addressed, explicit recognition of methodological constraints would improve transparency. Two key limitations should be acknowledged:

- a. Lack of genetic knockout validation: Pharmacological approaches provide valuable insights, but without genetic knockouts, off-target effects and compensatory mechanisms remain unaccounted for.
- b. Pleiotropic effects of serotonin modulation: Serotonin influences multiple neural pathways, and its role in regeneration may be indirect.

The pharmacological approach in this study is a strength, and I appreciate the authors' efforts in systematically testing serotonin receptor modulators. However, the argument that further characterization is "out of scope" is problematic because the mechanistic conclusions depend on these molecules. Claiming serotonin is required for regeneration without fully characterizing its mode of action is like asserting a drug is effective without determining how it works.

My critique does not mean that the authors need to conduct an exhaustive characterization or study extension. But at a minimum, the authors should clarify whether alternative serotonin-related

pathways have been ruled out or considered, as this affects interpretation of the data. They can also acknowledge that additional mechanistic validation (e.g., conditional receptor-specific genetic knockouts) would further strengthen their conclusions (best would be in a limitations section).

Reviewer 3

Advance summary and potential significance to field

As stated previously, the authors provide evidence, using 5-HT agonists and antagonists, for a specific role for serotonin signaling in guiding regenerating RGC axons after optic nerve injury. The authors establish that this role is specific to regenerating axons and is not essential for developmental RGC axon growth. The authors also demonstrate how their previously established larval zebrafish model for optic nerve injury can be utilized for drug screening to identify pharmaceuticals that impact the normally successful regeneration of CNS nerves in zebrafish. Both the finding for an *in vivo* role of 5-HT in modulating optic nerve regeneration, and the demonstration of the utility of the model for pharmaceutical screening are significant contributions to the field. The data presented support the conclusion of a role for 5-HT signaling in guidance of RGC axons to the chiasm during optic nerve regeneration and lend support to the notion that axon regeneration is not simply a recapitulation of developmental axon growth.

Comments for the author

The authors' responses to reviewers mostly met with my satisfaction. However, since a major contribution of this paper was demonstrating the utility of the larval optic nerve regeneration model for screening pharmaceuticals, a table of the 40 pathways they identified in their screen is necessary. The vague references to "growth factor pathways, neuronal cell surface receptors and neurotransmitter systems" are unsatisfactory. I understand if they are not ready to disclose the chemicals since they may be following those up in additional studies, however, not disclosing the pathways identified along with a rationale for selecting the serotonin signaling pathway is much needed.

Second revision

Author response to reviewers' comments

Reviewer 1: SUMMARY OF THE ADVANCE MADE IN THIS PAPER AND ITS POTENTIAL SIGNIFICANCE TO THE FIELD

This manuscript reports a previously uncharacterized function of serotonin type-1 receptors (5-HT_{1s}) in optic nerve (ON) regeneration *in vivo*. Using an elegant ON transection assay in zebrafish combined with a small molecule library screen, the authors identified several compounds that modulate serotonin signaling and modified ON regeneration. The authors then demonstrate that 5-HT₁ receptors are expressed by retinal ganglion cells (RGCs) before transection and during ON regeneration, and that inhibiting 5-HT_{1A} receptors reduces regeneration but does not affect developmental axon outgrowth. The authors finally show that activating 5-HT_{1B/D} receptors can enhance regeneration. Overall, this study makes a significant contribution to the field by uncovering a novel and specific function of 5-HT₁ receptors in ON regeneration.

SUGGESTIONS TO AUTHORS

The study is well carried out, technically of high standard, and provides a thoughtful discussion about the functions of serotonin signaling in the regulation of nerve regeneration. The additional experiments showing that some regenerating axons have reached the midline at 32 hpt and that *htr1aa/ab* receptors are no longer expressed at that timepoint reinforce the conclusions of the study.

-We want to thank the reviewer for their thoughtful comments and suggestions. Their reviews have helped us improve the quality of our manuscript.

Reviewer 2: SUMMARY OF THE ADVANCE MADE IN THIS PAPER AND ITS POTENTIAL SIGNIFICANCE TO THE FIELD

This study provides valuable insights into optic nerve regeneration by highlighting the role of serotonin signaling. The use of zebrafish as a model, coupled with pharmacological screening, contributes to understanding CNS regenerative mechanisms. The findings have potential significance in broader neuroregeneration research, particularly in CNS repair strategies. The authors have addressed several concerns.

SUGGESTIONS TO AUTHORS

That said, a few critical issues remain unresolved, and I would encourage the authors to engage with them directly to further enhance the clarity, rigor, and impact of their study.

Essential revisions:

1. statistics:

The omission of individual data points remains a significant concern, as it obscures variance and prevents a complete assessment of data distribution. This is not a subjective preference but a requirement stated in the journal's guidelines (Journal's checklist <https://journals.biologists.com/DocumentLibrary/DEV/Checklist.pdf> states clearly: Graphs should allow the reader to see the true data spread (i.e., box-and-whisker plots, SuperPlots, etc.). For small sample sizes, individual data points should be plotted.

Without individual data points, key statistical properties such as variance, skewness, kurtosis, and F-values cannot be fully assessed. Statistical summaries, while useful, are not a substitute for raw data visualization. Moreover, no raw data file is provided, making independent verification impossible. Since presenting individual data points is an explicit requirement of the journal, the authors should include them. If there is a compelling methodological reason for not doing so, this should be transparently stated rather than framed as a mere "choice of data presentation."

-We want to thank the reviewer for taking the time to read our 'Response to Reviewers' cover letter, our manuscript revisions, and for sharing their valuable feedback. We appreciate the discussion and suggestions brought by the reviewer about the appropriate statistics and data visualization of our data. Based on these suggestions, we decided to create additional stacked bar graphs for each optic nerve regeneration experiment included in Figures 2 and 5 (see an example of the new Supplemental Figure 1B below). We chose stacked bar graphs because our results come from categorical data. These graphs show the percentage of optic nerves with a given phenotype for control and experimental groups. They also show the total number of samples that fell into each category (in the example below, for the transected animals treated with 5 μ M of WAY-100635 (second bar graph from left to right), 22 optic nerves re-innervated the contralateral tectum by 48 hpt and 3 optic nerves did NOT re-innervated the contralateral tectum).

We are also happy to share the raw data of our results. We will include a spreadsheet as part of our Supplemental Material (titled Supplemental Table 2: Raw Data for Saied-Santiago et al.) that includes data and statistics for all experiments included in each main figure of our manuscript.

Figure S1B. Numbers inside the bars correspond to the number of optic nerves with a given phenotype. Please see the manuscript resubmission for the Figure Legend(s).

However, we would still prefer to keep our graphs using the normalized data showing the fold change difference between experimental and control groups as the main figures of the manuscript.

2. long-range

The distinction between "long-range" and "short-range" regeneration remains unclear. The term does not appear in the title or abstract, was not there in the initial submission, suggesting it was introduced post hoc.

If the authors classify optic nerve regeneration as "long-range," then what constitutes "short-range" in this context? Given that peripheral nerves and some spinal cord models regenerate over even longer distances, it is unclear why this term is relevant. Can the authors provide a biologically meaningful definition rather than using this term as a general rhetoric justification?

-We want to thank the reviewer for raising this important point about regeneration distance. We also want to apologize for the confusion caused by not highlighting changes we made to the discussion about short-range vs. long-range regeneration. After re-reading our manuscript, we agree that the 'long-range' term was used incorrectly and subjectively. Therefore, we decided to remove all instances in which 'long-range' or 'long-distance' was used in the manuscript. We believe that the statements written about optic nerve regeneration in our manuscript are supported without using these adjectives.

3. limitations

The study lacks a dedicated limitations section. While alternative explanations are addressed, explicit recognition of methodological constraints would improve transparency. Two key limitations should be acknowledged:

- Lack of genetic knockout validation: Pharmacological approaches provide valuable insights, but without genetic knockouts, off-target effects and compensatory mechanisms remain unaccounted for.
- Pleiotropic effects of serotonin modulation: Serotonin influences multiple neural pathways, and its role in regeneration may be indirect.

The pharmacological approach in this study is a strength, and I appreciate the authors' efforts in systematically testing serotonin receptor modulators. However, the argument that further characterization is "out of scope" is problematic because the mechanistic conclusions depend on these molecules. Claiming serotonin is required for regeneration without fully characterizing its mode of action is like asserting a drug is effective without determining how it works.

My critique does not mean that the authors need to conduct an exhaustive characterization or study extension. But at a minimum, the authors should clarify whether alternative serotonin-related pathways have been ruled out or considered, as this affects interpretation of the data. They can also

acknowledge that additional mechanistic validation (e.g., conditional receptor-specific genetic knockouts) would further strengthen their conclusions (best would be in a limitations section).

-We want to thank the reviewer for their valuable feedback on the Discussion section of the manuscript. Based on their suggestions, we will include a dedicated limitations section. We would like to clarify that additional small molecules that target the 5-HT signaling pathway were included in our Drug Screen. While treatment with these small molecules did NOT lead to a robust phenotype in optic nerve regeneration, there are several explanations (such as small sample size, drug concentration, and combination effect with other drugs in each drug pool) as to why these or any of the small molecules tested in the Drug Screen did not have an effect in optic nerve regeneration. Therefore, we are not able to rule out or discuss in detail any of the serotonin-related pathways targeted by small molecules with no phenotype in this process.

The last section of the Discussion will be titled "Conclusions and Limitations", and the first paragraph will emphasize the limitations of the study and will focus on the two key limitations mentioned by the reviewer (see section inside the quotes for new paragraph):

"While pharmacological approaches are a strength of our study, one limitation is that we cannot entirely exclude the possibility that these approaches may lead to some off-target effects in vivo. Moreover, due to the potential pleiotropic effects of 5-HT modulation in the CNS, 5-HT receptor signaling might indirectly promote optic nerve regrowth to the CNS midline indirectly through neuromodulation of yet to be defined guidance cues. This interpretation is supported by previous research elucidating the role of 5-HT1B/1D receptor signaling in mediating the response of thalamocortical axons to netrin (Bonnin et al., 2007) and 5-HT2 receptors directing the midline crossing of commissural axons by regulating the translation of ephrinB2 (Xing et al., 2015). Therefore, using conditional knock-outs and similar molecular methods to target specific 5-HT receptors is an important next step to further our understanding of 5-HT receptor signaling in CNS and optic nerve regeneration". The following paragraph of this section is a paragraph already part of the main text in the Discussion that reads: "Our findings are consistent with a model in which tight regulation of 5-HT1 receptor signaling levels is needed...". We will close the section with future work.

Reviewer 3: SUMMARY OF THE ADVANCE MADE IN THIS PAPER AND ITS POTENTIAL SIGNIFICANCE TO THE FIELD

As stated previously, the authors provide evidence, using 5-HT agonists and antagonists, for a specific role for serotonin signaling in guiding regenerating RGC axons after optic nerve injury. The authors establish that this role is specific to regenerating axons and is not essential for developmental RGC axon growth. The authors also demonstrate how their previously established larval zebrafish model for optic nerve injury can be utilized for drug screening to identify pharmaceuticals that impact the normally successful regeneration of CNS nerves in zebrafish. Both the finding for an in vivo role of 5-HT in modulating optic nerve regeneration, and the demonstration of the utility of the model for pharmaceutical screening are significant contributions to the field. The data presented support the conclusion of a role for 5-HT signaling in guidance of RGC axons to the chiasm during optic nerve regeneration and lend support to the notion that axon regeneration is not simply a recapitulation of developmental axon growth.

SUGGESTIONS TO AUTHORS

The authors' responses to reviewers mostly met with my satisfaction. However, since a major contribution of this paper was demonstrating the utility of the larval optic nerve regeneration model for screening pharmaceuticals, a table of the 40 pathways they identified in their screen is necessary. The vague references to "growth factor pathways, neuronal cell surface receptors and neurotransmitter systems" are unsatisfactory. I understand if they are not ready to disclose the chemicals since they may be following those up in additional studies, however, not disclosing the pathways identified along with a rationale for selecting the serotonin signaling pathway is much needed.

-We want to thank the reviewer for reading the author's responses and giving us additional feedback that will improve the final version of our manuscript. We apologize for not understanding their concern and the importance of sharing the molecular pathways analyzed and identified in our small molecule

screen as having a potential role in optic nerve regeneration. As suggested above, we created a table (S1 table in the Supplemental Materials section) containing the following columns: Pathways (as defined by the PANTHER database), Agonist/Antagonist (whether agonists and antagonists were analyzed for a given pathway), and Target (1-2 examples). We want to clarify that this list of pathways includes every analyzed pathway (i.e., some of these pathways did not have any 'hits' with a phenotype in optic nerve regeneration). In addition, we have highlighted five pathways in blue that represent the pathways with the highest number of small molecules 'hits'. Each small molecule 'hit' was re-tested to determine which small molecule and pathway would be analyzed in detail moving forward. Given that most of the small molecule 'hits' in the serotonin pathway were successfully validated (4 out of 6), and the known role of the 5-HT pathway in neuronal growth and guidance, we selected this pathway to further characterize its role in optic nerve regeneration.

Third decision letter

MS ID#: dev.204334R2

MS TITLE: Serotonin neuromodulation directs optic nerve regeneration

AUTHORS: Kristian Saied-Santiago; Melissa Baxter; Jaffna Mathiapparanam; Michael Granato

Dear Kristian and Michael,

I am happy to tell you that your manuscript has been accepted for publication in Development, pending our standard publication integrity checks.